# 3D-RFT: Reinforcement Fine-Tuning for Video-based 3D Scene Understanding

Xiongkun Linghu [1 * †]   Jiangyong Huang [1 2 *]   Baoxiong Jia [1]   Siyuan Huang [1]

## Abstract

Reinforcement Learning with Verifiable Rewards (RLVR) has emerged as a transformative paradigm for enhancing the reasoning capabilities of Large Language Models (LLMs), yet its potential in 3D scene understanding remains underexplored. Existing approaches largely rely on Supervised Fine-Tuning (SFT), where the token-level cross-entropy loss acts as an indirect proxy for optimization, leading to a misalignment between training objectives and task performances. To bridge this gap, we present 3D-RFT, a systematic framework to extend RLVR to video-based 3D perception and reasoning. 3D-RFT shifts the paradigm by directly optimizing the model towards evaluation metrics, performing reinforcement fine-tuning using Group Relative Policy Optimization (GRPO) with strictly verifiable reward functions, which directly stem from metrics like 3D IoU and F1-Score to provide more effective learning signals. Extensive experiments demonstrate that 3D-RFT-4B achieves state-of-the-art performance on various video-based 3D scene understanding tasks. Notably, 3D-RFT-4B significantly outperforms larger models (*e.g.*, VG LLM-8B) on 3D video detection, 3D visual grounding, and spatial reasoning benchmarks. We further showcase advantages of 3D-RFT, such as robust efficacy, and provide valuable insights into training strategies and data impact. We hope 3D-RFT can serve as a robust and promising paradigm for future development of 3D scene understanding. Code is available on project page.

[*]Equal contribution ; [†]Project lead. [1]State Key Laboratory of General Artificial Intelligence, BIGAI [2]Peking University. Correspondence to: Siyuan Huang <huangsiyuan@ucla.edu>, Baoxiong Jia <baoxiongjia@g.ucla.edu>.

*Proceedings of the 43 $^{rd}$ International Conference on Machine Learning*, Seoul, South Korea. PMLR 306, 2026. Copyright 2026 by the author(s).

## 1. Introduction

The remarkable success of Multi-modal Large Language Models (MLLMs) (Bai et al., 2025b;a; Hurst et al., 2024; Liu et al., 2023; Team et al., 2023; Comanici et al., 2025) has driven advancements in diverse fields, including 3D scene understanding (Huang et al., 2024b;a; Zhu et al., 2025a; Linghu et al., 2024; Zheng et al., 2025c;b; Chen et al., 2026), robotic manipulation (Driess et al., 2023; Brohan et al., 2023; Team et al., 2025a), and embodied navigation (Zheng et al., 2024; Zhang et al., 2024; Cheng et al., 2024a).

Treating 3D scenes as video streams offers a scalable alternative to traditional point-cloud and depth-based methods, bypassing the need for specialized sensors. By leveraging widely available RGB cameras and the temporal capabilities of MLLMs, this paradigm has become a central focus of recent research (Zheng et al., 2025c;b; Zhu et al., 2025a; Huang et al., 2025b; Fan et al., 2025). While some works (Zheng et al., 2025c; Zhu et al., 2025a) still inject explicit 3D features, others (Zheng et al., 2025b; Fan et al., 2025) extract 3D priors directly from video frames. Crucially, VG LLM (Zheng et al., 2025b) validates this approach on tasks like cross-frame detection and 3D visual grounding.

However, these approaches primarily rely on SFT as the learning paradigm, which imposes an inherent performance ceiling due to the limitations of imitating labels. Specifically, in 3D perception tasks, model responses consist of 3D bounding boxes represented as sequences of textual floating-point numbers. SFT optimizes these sequences by minimizing a per-token Cross-Entropy (CE) loss, which creates a critical misalignment: optimization occurs in the discrete token space, whereas evaluation is conducted in the continuous 3D coordinate system. Since the output tokens must be decoded and parsed into geometric structures to compute metrics like 3D Intersection over Union (IoU), the standard SFT objective acts only as an indirect proxy, failing to capture the underlying landscape of the predictions.

Incorporating metric-driven loss functions directly into SFT remains a key challenge due to the non-differentiable nature of evaluation pipelines. Because standard SFT relies on backpropagation, it requires a fully differentiable path from the loss to the model's logits. Using target 3D metrics as an SFT loss is mathematically intractable; converting text to a 3D box requires discrete string parsing, and calculat-

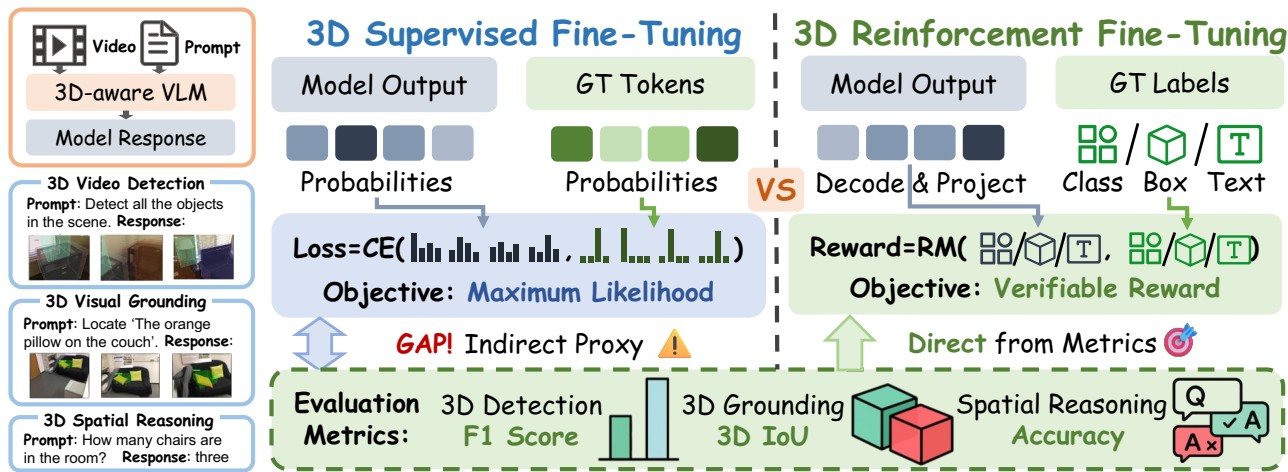

Figure 1. **Comparison between SFT and 3D-RFT training paradigms.** Left: Standard Supervised Fine-Tuning (SFT) relies on a per-token Cross-Entropy loss which acts as an indirect proxy, leading to a gap between training objectives and final evaluation metrics. Middle: Identical output formats for 3D Video Detection, Grounding, and Spatial Reasoning tasks. Right: 3D-RFT utilizes a Scalar Reward (Policy Gradient) derived directly from evaluation metrics (*e.g.*, F1-Score, 3D IoU, and Accuracy) through a decoding and parsing module, ensuring the model directly optimizes for the final task performance.

ing the final metric involves non-continuous step functions (e.g., $IoU > 0.25$). If integrated directly, these operations produce zero or undefined gradients, halting the learning process.

How can we overcome the limits of **answer-only supervision** and the **non-differentiable barrier** of SFT when applying metric-driven supervised signals? RLVR offers a principled solution by optimizing models directly against **Verifiable Rewards**, which are derived directly from distinct evaluation metrics or pre-defined rules. Unlike SFT, which constrains the model to mimic ground-truth sequences via a per-token CE loss, Reinforcement Learning (RL) optimizes the policy based on scalar reward scores. This provides a superior optimization target that is strictly aligned with the evaluation process, encouraging sequence exploration without the limitations of token-level penalties. Recent breakthroughs, such as GPT-o1 (Jaech et al., 2024b) and DeepSeek-R1 (Guo et al., 2025), have widely demonstrated the success of RLVR in advancing mathematical reasoning and code generation.

A natural question arises: *Can this metrics-driven RL paradigm generalize to video-based 3D scene understanding?* To address this, we introduce 3D-RFT, a unified framework that successfully extends RL to diverse video-based 3D scene understanding tasks, encompassing both *3D perception* and *3D spatial reasoning*.

As illustrated in Fig. 1, 3D-RFT fundamentally shifts the training objective. While SFT relies on an indirect proxy—minimizing CE loss between predicted and ground-truth probabilities—this leads to a misalignment between training objectives and evaluation metrics. In contrast, 3D-RFT leverages policy gradients derived directly from verifi-

able reward signals (*e.g.*, 3D IoU, Accuracy), ensuring the model is optimized explicitly for final task performance.

To overcome the lack of native 3D perception capabilities in existing MLLMs, we design a robust two-stage training pipeline: **1) SFT Warm-Up:** We first inject fundamental 3D awareness and scene understanding capabilities into the MLLM via SFT, establishing a stable policy initialization. **2) RL Training:** The model is then fine-tuned using the GRPO algorithm with verifiable reward functions strictly following evaluation protocols. For instance, in 3D Visual Grounding, we parse the predicted 9-DoF box, project it into the global coordinate system, and compute the global 3D IoU as the reward signal. This approach effectively transitions the learning paradigm from task-agnostic sequence imitation to metrics-driven policy optimization.

We validate the efficacy of 3D-RFT across typical video-based 3D scene understanding tasks, including 3D perception (*e.g.*, 3D video detection (Wang et al., 2024), 3D visual grounding (Chen et al., 2020)) and 3D spatial reasoning (Yang et al., 2025a). Experimental results demonstrate that 3D-RFT consistently enhances model performance, and our model 3D-RFT-4B outperforms larger baseline models. Our analysis yields several key findings: 1) 3D-RFT provides consistent gains across all three task domains. 2) In 3D perception tasks, 3D-RFT significantly boosts performance over the SFT baseline, surpassing even 8B-scale fine-tuned models on nearly all metrics, proving that metrics-driven optimization offers a more effective learning objective. 3) For 3D spatial reasoning, 3D-RFT effectively enhances model performance to state-of-the-art on VSI-Bench, surpassing previous models at larger scales; we also reveal the effects of data diversity on 3D-RFT.

In summary, our key contributions are as follows:

- We propose 3D-RFT, a systematic reinforcement fine-tuning framework that extends RLVR to 3D video perception and reasoning, shifting the learning paradigm from sequence imitation to metrics-driven policy optimization.
- We design task-specific verifiable reward functions derived directly from evaluation metrics (*e.g.*, 3D IoU, F1-Score) to enable efficient and robust policy updates.
- We conduct extensive experiments on standard video-based 3D scene understanding benchmarks, demonstrating that 3D-RFT achieves significant improvements over SFT baselines and surpasses larger-scale models in both 3D perception and reasoning tasks.

## 2. Related Work

**MLLMs for 3D Scene Understanding.** 3D perception and spatial reasoning are fundamental pillars of 3D scene understanding. Early research focused on developing unified MLLMs based on 3D sensory inputs (*e.g.*, point clouds) (Zhu et al., 2023; Xu et al., 2024; Huang et al., 2024b; Yang et al., 2024; Chen et al., 2024b; Qi et al., 2024; Zhu et al., 2024; Linghu et al., 2024; Fu et al., 2025; Chu et al., 2024; Deng et al., 2025; Mao et al., 2025). Recently, advancements in video understanding establish video-based MLLMs as a compelling approach due to the streamlined input and strong performance (El Banani et al., 2024; Man et al., 2024; Zhang et al., 2025c; Zhu et al., 2025a; Zheng et al., 2025c; Qi et al., 2025a; Huang et al., 2025b; Zhu et al., 2025b; Li et al., 2025b; Cheng et al., 2025). To bolster the 3D awareness of video-based representation, there are efforts in geometry enhancement (Wang et al., 2025a; Huang et al., 2025e; Zheng et al., 2025e; Hu et al., 2025) and data scaling (Chen et al., 2024a; Cheng et al., 2024b; Cai et al., 2025a; Ma et al., 2025b; Xu et al., 2025; Song et al., 2025; Zhang et al., 2025a; Zhou et al., 2025; Brown et al., 2025; Yang et al., 2025d; Cai et al., 2025b). However, these efforts are insufficient to resolve the persistent bottlenecks in 3D perception and spatial reasoning (Fu et al., 2024; Huang et al., 2025a; Yu et al., 2025b). In contrast, we contend that more critical issues reside in the SFT paradigm, *e.g.*, suboptimal objectives for perception tasks and overfitting issues for reasoning tasks. We address these bottlenecks through our meticulously designed 3D-RFT framework.

**Reinforcement Learning for MLLMs.** RLVR has catalyzed significant breakthroughs in reasoning capabilities of LLMs (Jaech et al., 2024a; Shao et al., 2024b; Guo et al., 2025; Yu et al., 2025a; Zheng et al., 2025a; Liu et al., 2025a;b; Yue et al., 2025; Wen et al., 2025) and MLLMs (Huang et al., 2025d; Liu et al., 2025c; Sarch et al., 2025; Tan et al., 2025). Recent efforts in video-based models have advanced Chain-of-Thought (CoT) data generation (Fei et al., 2024; Wu et al., 2024; Shao et al., 2024a; Linghu

et al., 2026; Chen et al., 2025a; Zheng et al., 2025d) and employed RLVR for video understanding (Feng et al., 2025; Li et al., 2025c; Wang et al., 2025c), 3D perception (Yuan et al., 2025b; Huang et al., 2025c; Wang et al., 2025d; Ma et al., 2025a), spatial reasoning (Zhan et al., 2025; Chen et al., 2025b; Wu et al., 2025a;b), and embodied intelligence (Qi et al., 2025b; Zhao et al., 2025; Yuan et al., 2025a). However, the efficacy of RLVR for 3D scene understanding tasks is under-explored (Liao et al., 2025; Zhong et al., 2025). A concurrent work, VST (Yang et al., 2025b), explores RLVR for training a generalist video model on various spatial tasks. In contrast, we present a systematic study of RLVR spanning 3D perception, spatial-temporal grounding, and spatial reasoning, offering insights into learning objectives, model components, data configurations, and training dynamics.

## 3. Methodology

### 3.1. Preliminary

**Reinforcement Learning with Verifiable Rewards (RLVR).** RLVR has emerged as a powerful paradigm for enhancing reasoning in LLMs (Guo et al., 2025; Team et al., 2025b). Unlike SFT that maximizes the likelihood of expert data, RLVR maximizes the expected value of a verifiable outcome. Given an input prompt $\mathbf{x}$, the policy $\pi_\theta$ generates a response $\mathbf{y}$, which is evaluated by a deterministic verifier $\mathcal{V}$ to yield a reward $R = \mathcal{V}(\mathbf{x}, \mathbf{y})$. The objective is to maximize the expected reward over the data distribution:

$$J(\theta) = \mathbb{E}_{\mathbf{x} \sim \mathcal{D}, \mathbf{y} \sim \pi_\theta(\cdot|\mathbf{x})}[R]. \tag{1}$$

**Group Relative Policy Optimization (GRPO).** GRPO (Shao et al., 2024b) is a memory-efficient variant of PPO (Schulman et al., 2017) that eliminates the need for a separate critic network. Instead of learning a value function to estimate the baseline, GRPO samples a group of outputs $\{\mathbf{y_1}, \ldots, \mathbf{y_G}\}$ for each prompt $\mathbf{x}$ from the old policy $\pi_{\theta_{old}}$. The advantage $A_i$ for the $i$-th sample is then computed by normalizing its reward against the group statistics:

$$A_i = \frac{R_i - \text{mean}(\{R_1, \ldots, R_G\})}{\text{std}(\{R_1, \ldots, R_G\})}, \tag{2}$$

where $G$ is the group size. This approach significantly reduces memory overhead by removing the value model.

### 3.2. Task Formulation

We formulate video-based 3D scene understanding as a conditional text generation problem. Given the visual input $\mathbf{I}$ (composed of RGB video frames) and a textual query $\mathbf{x}$, the model generates a response sequence $\mathbf{y}$. To facilitate verifiable reward computation, we enforce a structured output format: the model must first generate a reasoning chain enclosed in `<think>` tags, followed by the final prediction within `<answer>` tags. We consider two perception

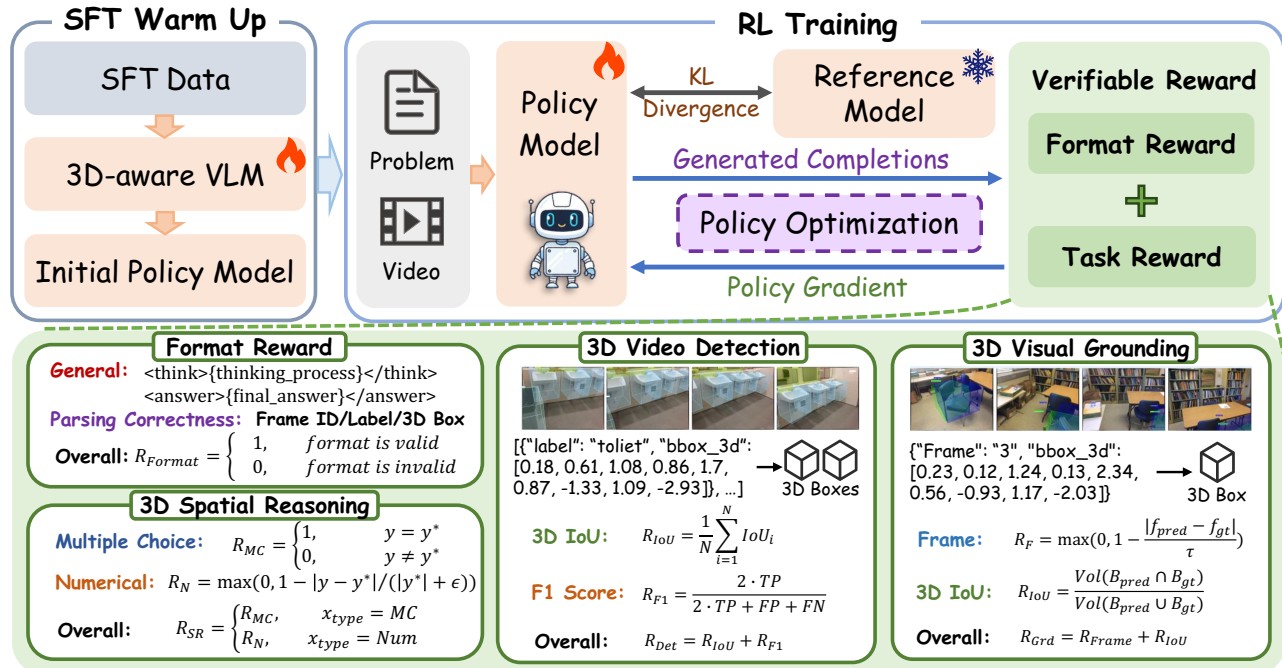

Figure 2. **Overview of the 3D-RFT training framework.** The process consists of two main stages: (1) **SFT Warm Up**: Initial training of the 3D-aware VLM using SFT data to establish a baseline policy. (2) **RL Training**: The Policy Model generates completions for video-based problems. A Verifiable Reward is calculated based on **Format Reward** (adherence to structured output) and **Task Reward** (performance in 3D Video Detection, 3D Visual Grounding, and Spatial Reasoning using metrics like 3D IoU and F1 Score). The model is optimized via Policy Gradient while maintaining a KL Divergence constraint relative to the frozen Reference Model.

tasks (3D video detection and 3D visual grounding) and a reasoning task (3D spatial reasoning). For 3D perception tasks, the final answer is formatted as a JSON object where 3D bounding boxes are represented as 9-DoF tuples: $\mathbf{b} = (x, y, z, w, h, d, \psi, \theta, \phi)$.

**3D Video Detection.** The goal is to detect all objects $\mathcal{O} = \{(\mathbf{b_i}, c_i)\}$ appearing throughout the image frames, where $\mathbf{b_i}$ refers to the 9-DoF bounding box and $c_i$ refers to the object category. All the bounding boxes are unified into the **coordinate system of the first frame**.

**3D Visual Grounding.** Given a grounding text, the model predicts a tuple $(f_{\text{pred}}, \mathbf{b}_{\text{pred}})$, identifying both the frame index and the object's 9-DoF bounding box. Unlike detection, $\mathbf{b}_{\text{pred}}$ is defined in the **local coordinate system of the predicted frame** $f_{\text{pred}}$.

**3D Spatial Reasoning.** Given a question, the model must generate accurate text answers. This task mainly focuses on spatial attributes and relations within 3D scenes.

### 3.3. Training Pipeline

As illustrated in Figure 2, our training pipeline consists of two stages: (1) **SFT Warm Up**, which activates the model's capabilities of 3D scene understanding and establishes an initial policy model; and (2) **RL Training**, where the model

is optimized using strictly verifiable rewards. This second stage leverages policy gradients derived from both format constraints and task-specific metrics (*e.g.*, 3D IoU) to directly refine the model's performance on video-based 3D scene understanding tasks.

#### 3.3.1. STAGE 1: SFT WARM UP

Before RL training, the model must learn to follow the required output format (*i.e.*, enclosing reasoning in `<think>` tags and presenting the final answer in proper format) and acquire basic capabilities to provide a good starting point for the subsequent RL stage. In the SFT stage, the model is trained to maximize the log-likelihood of the ground-truth response $\mathbf{y}^*$ given the visual input $\mathbf{I}$ and textual query $\mathbf{x}$:

$$\mathcal{L}_{\text{SFT}}(\theta) = -\sum_{t=1}^{T} \log \pi_\theta(\mathbf{y_t^*} \mid \mathbf{x}, \mathbf{I}, \mathbf{y_{<t}^*}). \quad (3)$$

#### 3.3.2. STAGE 2: RL TRAINING

In the second stage, we proceed to refine the model by employing GRPO with verifiable rewards. For each visual-textual input $(\mathbf{I}, \mathbf{x})$, we sample a group of $G$ outputs $\{\mathbf{y_1}, \ldots, \mathbf{y_G}\}$ from the current policy $\pi_{\theta_{\text{old}}}$. The advantage $A_i$ for each sample is computed by normalizing the rewards within the group. The objective maximizes the expected advantage while constraining the policy shift via a

KL-divergence penalty with the reference model $\pi_{\text{ref}}$:

$$\mathcal{L}_{\text{GRPO}}(\theta) = -\frac{1}{\sum_{i=1}^{G} T_i} \sum_{i=1}^{G} \sum_{t=1}^{T_i} \mathcal{L}_{i,t}(\theta) + \beta \mathbb{D}_{\text{KL}}[\pi_\theta || \pi_{\text{ref}}], \quad (4)$$

$$\mathcal{L}_{i,t}(\theta) = \min\left(r_{i,t} A_i, \text{clip}(r_{i,t}, 1 - \epsilon, 1 + \epsilon) A_i\right), \quad (5)$$

$$r_{i,t} = \frac{\pi_\theta(\mathbf{y_{i,t}}|\mathbf{I_i}, \mathbf{x_i}, \mathbf{y_{i,<t}})}{\pi_{\theta_{\text{old}}}(\mathbf{y_{i,t}}|\mathbf{I_i}, \mathbf{x_i}, \mathbf{y_{i,<t}})}. \quad (6)$$

The advantage $A_i$ is defined in Equation (2). To mitigate the high memory cost of long video contexts during training, we adopt a loss chunking technique (see details in Sec. A.2).

### 3.4. Verifiable Reward Design

As shown in Figure 2, our rewards include two components: a **Format Reward** to enforce structural validity (*e.g.*, correct JSON syntax and bounding box tuples), and **Task Rewards** that directly optimize geometric and semantic metrics. We will elaborate on **Task Rewards** as follows.

#### 3.4.1. 3D VIDEO DETECTION: 3D IOU AND F1-SCORE

In the 3D video detection task, the model outputs a set of predicted bounding boxes $\mathcal{B}_{\text{pred}} = \{\mathbf{b_1}, \ldots, \mathbf{b_N}\}$. Each predicted bounding box $\mathbf{b_i}$ is directly compared against the ground-truth labels $\mathcal{B}_{\text{gt}}$ to derive the maximum 3D IoU $\mathcal{I}_i$. We adopt **Average IoU Reward** to provide a dense learning signal by averaging the maximum IoUs across all predictions:

$$R_{\text{IoU}}^{(\text{Det})} = \frac{1}{N} \sum_{i=1}^{N} \mathcal{I}_i. \quad (7)$$

To further enhance detection capability, we introduce the **F1-Score Reward**. A prediction is counted as a True Positive (TP) if it matches an unmatched ground-truth bounding box with a 3D IoU exceeding $\tau_{\text{F1}}$ (set to 0.25 by default); otherwise, it is treated as a False Positive (FP). Ground-truth bounding boxes that remain unmatched are regarded as False Negatives (FN). The F1-Score Reward is then computed as:

$$R_{\text{F1}} = \frac{2 \cdot \text{TP}}{2 \cdot \text{TP} + \text{FP} + \text{FN}}. \quad (8)$$

The **Average IoU Reward** ensures valid geometric precision, while the **F1-Score Reward** directly optimizes the final evaluation metric. The combined verifiable reward for 3D video detection is:

$$R_{\text{Det}} = R_{\text{IoU}}^{(\text{Det})} + R_{\text{F1}}. \quad (9)$$

#### 3.4.2. 3D VISUAL GROUNDING: FRAME AND 3D IOU

Video-based 3D visual grounding requires spatio-temporal localization of a specific object according to the grounding text. We consider both temporal and spatial precision.

**Temporal Reward.** Locating the target frame index $f_{\text{gt}}$ is formulated as a regression task. To provide a dense optimization signal, we employ a smoothed linear decay function based on the absolute temporal distance between the predicted frame index $f_{\text{pred}}$ and the ground truth, governed by a tolerance threshold $\tau_{\text{frame}}$ (set to 5 by default):

$$R_{\text{frame}} = \max\left(0, 1 - \frac{|f_{\text{pred}} - f_{\text{gt}}|}{\tau_{\text{frame}}}\right). \quad (10)$$

**IoU Reward.** To align with the evaluation metric, we compute the 3D IoU in the global coordinate system. Given the predicted bounding box $\mathbf{b}_{\text{pred}}$ in the local camera coordinates of frame $f_{\text{pred}}$, we first transform it to the global scene coordinates using the frame's extrinsic matrix $M_{c \to g}^{(f_{\text{pred}})}$ and the scene's axis-alignment matrix $M_{\text{align}}$: $\mathbf{b}'_{\text{pred}} = M_{\text{align}} M_{c \to g}^{(f_{\text{pred}})} \mathbf{b}_{\text{pred}}$. The IoU reward is then calculated by standard 3D IoU between the aligned prediction and the ground-truth box $\mathbf{b}_{\text{gt}}$:

$$R_{\text{IoU}}^{(\text{Grd})} = \frac{|\mathbf{b}'_{\text{pred}} \cap \mathbf{b}_{\text{gt}}|}{|\mathbf{b}'_{\text{pred}} \cup \mathbf{b}_{\text{gt}}|}. \quad (11)$$

The final reward for 3D visual grounding is defined as:

$$R_{\text{Grd}} = R_{\text{frame}} + R_{\text{IoU}}^{(\text{Grd})}. \quad (12)$$

#### 3.4.3. 3D SPATIAL REASONING: ACCURACY

For the 3D spatial reasoning task, we design the accuracy reward $R_{\text{acc}}$ by dispatching task-specific verifiers based on the question type:

- **Multiple Choice.** For multi-choice scenarios, we use an exact-match indicator function:

$$R_{\text{MC}} = \mathbb{1}(\mathbf{y} = \mathbf{y}^*) \quad (13)$$

- **Numerical Reasoning.** For numerical answers (*e.g.*, counting), we employ the Mean Relative Accuracy (MRA) with $\mathcal{C} = \{0.50, 0.55, \ldots, 0.95\}$ (Yang et al., 2025a):

$$R_{\text{num}} = \frac{1}{10} \sum_{\tau_{\text{num}} \in \mathcal{C}} \mathbb{1}\left(\frac{|\mathbf{y} - \mathbf{y}^*|}{|\mathbf{y}^*|} < 1 - \tau_{\text{num}}\right). \quad (14)$$

## 4. Experiments

In this section, we experiment with applying 3D-RFT to 3D perception and spatial reasoning tasks. We present the results and analyses of 3D perception tasks in Sec. 4.1, and 3D spatial reasoning task in Sec. 4.2. Additionally, we present the training dynamics of both tasks in Sec. 4.3 to visualize the model's behaviors during 3D-RFT.

*Table 1.* **Quantitative results on ScanNetDetection.** In this table, we present the comparison with baseline models and report the performance improvement between the SFT baseline (VG LLM-4B) and 3D-RFT-4B (ours).

| Model | chair | cabinet | table | bin | couch | bed | bathtub | toilet | 20 Common Classes | | |
| --- | --- | --- | --- | --- | --- | --- | --- | --- | --- | --- | --- |
| | | | | | | | | | $P_{25}$ | $R_{25}$ | $F1_{25}$ |
| *4-Frame Setting* | | | | | | | | | | | |
| Qwen2.5-VL-3B (Zheng et al., 2025b) | 37.7 | 10.2 | 35.0 | 23.1 | 39.0 | 64.8 | 32.4 | 68.8 | 32.6 | 27.9 | 30.0 |
| Qwen2.5-VL-7B (Zheng et al., 2025b) | 41.2 | 11.6 | 36.5 | 30.2 | 41.1 | 68.2 | 36.6 | 68.7 | 34.6 | 31.0 | 32.5 |
| VG LLM-4B (Zheng et al., 2025b) | 49.7 | 13.1 | 41.3 | 39.2 | 44.6 | 71.2 | 33.5 | 83.4 | 41.7 | 35.7 | 38.2 |
| VG LLM-8B (Zheng et al., 2025b) | 54.0 | 17.1 | 46.5 | **39.8** | 47.0 | 74.1 | 42.1 | 82.5 | 43.4 | **39.6** | 41.2 |
| 3D-RFT-4B (ours) | **55.3** | **18.7** | **48.2** | 39.1 | **49.8** | **77.1** | **50.0** | **86.1** | **54.2** | 38.2 | **43.7** |
| *Improvement* | +5.6 | +5.6 | +6.9 | -0.1 | +5.2 | +5.9 | +16.5 | +2.7 | +12.5 | +2.5 | +5.5 |
| *6-Frame Setting* | | | | | | | | | | | |
| Qwen2.5-VL-3B (Zheng et al., 2025b) | 32.8 | 7.8 | 31.3 | 20.9 | 32.2 | 58.8 | 36.5 | 66.1 | 27.8 | 24.1 | 25.7 |
| Qwen2.5-VL-7B (Zheng et al., 2025b) | 36.1 | 10.6 | 32.7 | 25.0 | 40.7 | 64.6 | 38.4 | 68.6 | 31.8 | 28.0 | 29.6 |
| VG LLM-4B (Zheng et al., 2025b) | 41.6 | 12.4 | 39.8 | 33.1 | 45.0 | 70.2 | 33.8 | 80.6 | 39.7 | 34.0 | 36.4 |
| VG LLM-8B (Zheng et al., 2025b) | 48.7 | 17.9 | 44.8 | **38.5** | 46.4 | **75.8** | 40.4 | **83.2** | 43.5 | **38.7** | 40.8 |
| 3D-RFT-4B (ours) | **49.2** | **18.5** | 44.9 | 34.6 | **48.3** | 74.1 | **51.4** | 81.1 | **53.4** | 35.7 | **41.7** |
| *Improvement* | +7.6 | +6.1 | +5.1 | +1.5 | +3.3 | +3.9 | +17.6 | +0.5 | +13.7 | +1.7 | +5.3 |

*Table 2.* **Quantitative results on ScanRefer.** The content in "()" indicates results with proposal refinement (Zheng et al., 2025b).

| Model | 3D Scene Input | Acc@0.25 | Acc@0.5 |
| --- | --- | --- | --- |
| ScanRefer (Chen et al., 2020) | ✓ | 37.3 | 24.3 |
| MVT (Huang et al., 2022b) | ✓ | 40.8 | 33.3 |
| ViL3DRel (Chen et al., 2022) | ✓ | 47.9 | 37.7 |
| 3D-LLM (Chen et al., 2024c) | ✓ | 30.3 | - |
| Chat-3D v2 (Huang et al., 2024a) | ✓ | 35.9 | 30.4 |
| Grounded 3D-LLM (Chen et al., 2024c) | ✓ | 47.9 | 44.1 |
| ChatScene (Huang et al., 2024a) | ✓ | 55.5 | 50.2 |
| LLaVA-3D (Zhu et al., 2025a) | ✓ | 54.1 | 42.4 |
| Video-3D LLM (Zheng et al., 2025c) | ✓ | **58.1** | **51.7** |
| SPAR (Zhang et al., 2025a) | ✗ | 31.9 (48.8) | 12.4 (43.1) |
| VG LLM-4B (Zheng et al., 2025b) | ✗ | 36.4 (53.5) | 11.8 (47.5) |
| VG LLM-8B (Zheng et al., 2025b) | ✗ | 41.6 (57.6) | 14.9 (50.9) |
| 3D-RFT-4B (ours) | ✗ | **42.9 (54.6)** | **15.9 (48.1)** |
| *Improvement* | ✗ | +6.5 (+1.1) | +4.1 (+0.6) |

*Table 3.* **Ablation study of training strategies and 3D priors.** We report accuracy at IoU thresholds 0.25 and 0.5.

| Training Strategy | 3D Prior | ScanRefer | |
| --- | --- | --- | --- |
| | | Acc@0.25 | Acc@0.5 |
| SFT | None | 31.9 (49.9) | 9.3 (43.8) |
| SFT → SFT | None | 34.2 (50.6) | 10.4 (44.9) |
| SFT → RL | None | 38.2 (52.7) | 12.1 (46.6) |
| SFT | VGGT | 36.4 (53.5) | 11.8 (47.5) |
| SFT → RL | VGGT | **42.9 (54.6)** | **15.9 (48.1)** |

**Comparison Baselines and Evaluation Metrics.** For 3D perception tasks, we mainly compare our model (*i.e.*, 3D-RFT-4B) with VG LLM (Zheng et al., 2025b) to demonstrate the effectiveness of 3D-RFT. Our evaluation metrics follow VG LLM: for ScanRefer, we report the accuracy at IoU thresholds of 0.25 and 0.5; for ScanNetDetection, we report precision, recall, and F1-score for 20 common object classes at an IoU threshold of 0.25. We use lmms_eval (Zhang et al., 2025b) for evaluation and keep the same evaluation hyper-parameters as VG LLM.

**Model.** We build our model 3D-RFT-4B based on VG LLM-4B (Zheng et al., 2025b), which consists of an MLLM backbone, Qwen2.5-VL-3B-Instruct (Bai et al., 2025b), and a visual geometry backbone VGGT-1B (Wang et al., 2025b). Features extracted from the VGGT backbone are first processed to align with the Qwen visual feature structure, and then combined via element-wise addition, producing a hybrid visual representation that is fed into the Qwen LLM.

**Results.** We report the detailed evaluation results of 3D video detection and 3D visual grounding in Tabs. 1 and 2, respectively. The results show that 3D-RFT-4B achieves state-of-the-art performances on 3D perception tasks. We provide further interpretations and analyses as follows.

### 4.1. 3D Perception Tasks

**Training Datasets.** For 3D perception tasks, we follow the settings of VG LLM (Zheng et al., 2025b), utilizing ScanRefer (Chen et al., 2020), Scan2Cap (Chen et al., 2021), and ScanNetDetection (Wang et al., 2024) for the SFT stage. We have not included CoT data in this stage due to the difficulty of acquiring high-quality data. In the second stage, we conduct Reinforcement Fine-Tuning (RFT) using ScanNetDetection and ScanRefer for 3D video detection and 3D visual grounding, respectively. We provide a detailed description of the data in Sec. A.1.

#### 4.1.1. 3D VIDEO DETECTION

**3D-RFT significantly enhances detection performance over the SFT baseline.** As detailed in Tab. 1, 3D-RFT-4B achieves substantial gains across all metrics. Under the *4-frame setting*, our method improves Precision by +12.5%, Recall by +2.5%, and F1-Score by +5.5% compared to the SFT baseline (VG LLM-4B). The improvements are robust across settings, with the *6-frame setting* yielding even higher gains in Precision (+13.7%) and comparable boosts in F1

(+5.3%). Notably, the improvements are most pronounced for larger objects, such as "bathtub" (+16.5%) and "table" (+6.9%), whereas smaller objects like "bin" show more limited gains, suggesting that further increases in visual resolution could potentially benefit small-object detection.

**3D-RFT-4B surpasses the larger VG LLM-8B model.** With only half the parameters, 3D-RFT-4B consistently outperforms VG LLM-8B on F1-Score and Precision for both *4-frame* and *6-frame* settings. Our model also achieves superior per-class performance on the majority of object categories. This demonstrates that 3D-RFT's metrics-driven optimization exploits the model's potential far more effectively than the standard SFT, enabling a 4B model to exceed the capabilities of an 8B baseline.

### 4.1.2. 3D VISUAL GROUNDING

**3D-RFT significantly enhances performance compared to the SFT baseline.** As shown in Tab. 2, 3D-RFT-4B achieves substantial gains over the VG LLM-4B baseline, elevating the Acc@IoU0.25 and Acc@IoU0.5 by +6.5% and +4.5% respectively. This performance leap confirms that the RFT stage effectively refines the policy's spatio-temporal grounding capabilities.

**3D-RFT-4B outperforms the larger VG LLM-8B.** With only half the parameters, 3D-RFT-4B surpasses VG LLM-8B (42.9% *vs.* 41.6% on Acc@IoU0.25). We attribute the advantage to the learning objective: while SFT focuses on maximizing the likelihood of ground-truth labels, 3D-RFT directly optimizes towards more geometrically precise grounding results. This effectively drives the model to learn tighter and more accurate bounding boxes, which is unlikely to emerge from the imitation learning of SFT.

### 4.1.3. ANALYSIS OF RFT AND 3D PRIOR

**The efficacy of RFT is robust across diverse visual inputs.** We further ablate the VGGT input to investigate whether the efficacy of 3D-RFT is consistent: (1) vanilla Qwen2.5-VL (No 3D Prior), and (2) Qwen2.5-VL augmented with VGGT. The results demonstrate consistent improvements under both settings, with the VGGT-equipped model improving from 36.4% to 42.9% on Acc@0.25 after RFT. This confirms that 3D-RFT consistently enhances the model's capability regardless of visual inputs, *e.g.*, the presence of 3D priors.

### 4.2. 3D Spatial Reasoning

**Formulation.** We consider two settings for the spatial reasoning task: (1) Direct-Answer (DA), where the model is prompted to directly output the final answer; and (2) Think-Answer (TA), where the model is prompted to think before giving the final answer. In this task, we conduct RFT in the TA setting following common practices, and

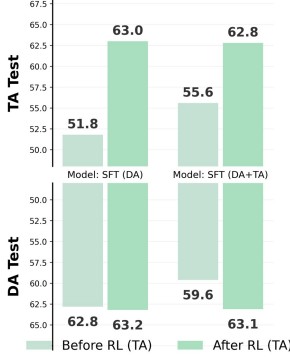
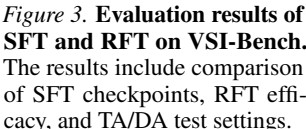
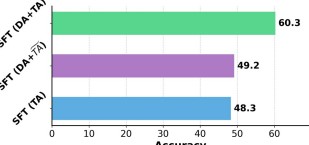

*Figure 4.* **VSI-Bench accuracy after 800 steps of RFT.** $\widetilde{TA}$ denotes lower-quality TA data.

*Figure 3.* **Evaluation results of SFT and RFT on VSI-Bench.** The results include comparison of SFT checkpoints, RFT efficacy, and TA/DA test settings.

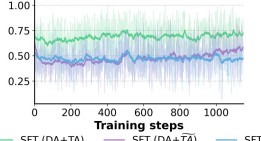

*Figure 5.* **Accuracy rewards during RFT.** $\widetilde{TA}$ denotes lower-quality TA data.

experiment with two schemes for the SFT stage: (1) default SFT (DA+TA), where we use both DA and TA data; and (2) SFT (DA), where we exclude TA data (*i.e.*, CoT data).

**Training Data.** We curate a data mixture from VLM-3R (Fan et al., 2025) and SpaceR (Ouyang et al., 2025), including 298K data instances in total, termed VSI-298K. Additionally, we generate 10K high-quality CoT data using Qwen3-VL-32B-Thinking (Bai et al., 2025a), termed CoT-10K. By default, we use the mixture of VSI-298K (DA) and CoT-10K (TA) for SFT, and VSI-298K for RFT (TA).

**Results.** We present the evaluation results on VSI-Bench in Tab. 4. The results show that 3D-RFT-4B outperforms prior methods by a large margin, especially on numerical reasoning categories. This suggests that RFT provides more effective learning signals compared to SFT. We provide additional results on more benchmarks in Sec. B.1, and further analyses on RFT efficacy and data as follows.

### 4.2.1. ANALYSIS OF RFT AND DATA

**3D-RFT consistently improves spatial reasoning performance.** As shown in Fig. 3, we present evaluation results regarding two SFT checkpoints, and compare their performances between "before" and "after" RFT. The results demonstrate that RFT yields consistent improvements on VSI-Bench, especially on TA test. Moreover, RFT on TA task elicits improvements on DA task. In addition, we observe that continually training with SFT yields a moderate performance drop, which suggests the advantage of RFT.

**DA data and TA data shape the foundation of RFT.** We compare the RFT results given different SFT data schemes: (1) default SFT (DA+TA); (2) SFT (DA+$\widetilde{TA}$), where we replace the CoT-10K data with 10K lower-quality CoT data generated by Qwen2.5-VL-72B-Instruct (Bai et al., 2025b); and (3) SFT (DA), which only uses CoT-10K during SFT. We compare their evaluation results and training rewards in

*Table 4.* **Quantitative results on VSI-Bench.** We include zero-shot results of base models, and test results of models after SFT and RFT.

| Model | Avg. | Numerical Answer | | | | Multiple-Choice Answer | | | |
| --- | --- | --- | --- | --- | --- | --- | --- | --- | --- |
| | | Obj. Count | Abs. Dist. | Obj. Size | Room Size | Rel. Dist. | Rel. Dir. | Route Plan | Appr. Order |
| *Base Models* | | | | | | | | | |
| GPT-4o (Hurst et al., 2024) | 34.0 | 46.2 | 5.3 | 43.8 | 38.2 | 37.0 | 41.3 | 31.5 | 28.5 |
| Gemini-2.5 Pro (Comanici et al., 2025) | 51.5 | 43.8 | 34.9 | 64.3 | 42.8 | 61.1 | 47.8 | **45.9** | 71.3 |
| LLaVA-Video-7B (Zhang et al., 2025c) | 35.6 | 48.5 | 14.0 | 47.8 | 24.2 | 43.5 | 42.4 | 34.0 | 30.6 |
| Qwen2.5-VL-7B (Bai et al., 2025b) | 32.7 | 34.5 | 19.4 | 47.6 | 40.8 | 32.8 | 24.5 | 32.5 | 29.4 |
| InternVL3-8B (Zhu et al., 2025c) | 42.1 | 68.1 | 39.0 | 48.4 | 33.6 | 48.3 | 36.4 | 27.3 | 35.4 |
| *Supervised Fine-Tuned* | | | | | | | | | |
| VG LLM-4B (Zheng et al., 2025b) | 47.3 | 66.0 | 37.8 | 55.2 | 59.2 | 44.6 | 45.6 | 33.5 | 36.4 |
| VLM-3R-7B (Fan et al., 2025) | 60.9 | 70.2 | 49.4 | 69.2 | **67.1** | **65.4** | **80.5** | 45.4 | 40.1 |
| Cambrian-S-3B (Yang et al., 2025d) | 57.3 | 70.7 | 40.6 | 68.0 | 46.3 | 64.8 | 61.9 | 27.3 | **78.8** |
| VST-SFT-3B (Yang et al., 2025b) | 57.9 | 69.3 | 45.4 | 71.8 | 62.4 | 59.0 | 46.0 | 38.7 | 70.2 |
| *Reinforcement Fine-Tuned* | | | | | | | | | |
| vsGRPO-2B (Liao et al., 2025) | 35.4 | 53.6 | 29.0 | 52.7 | 43.4 | 28.1 | 30.9 | 26.8 | 18.9 |
| vsGRPO-7B (Liao et al., 2025) | 40.7 | 59.9 | 29.6 | 50.8 | 48.3 | 35.4 | 35.6 | 34.0 | 31.5 |
| SpaceR-7B (Ouyang et al., 2025) | 43.5 | 61.9 | 28.6 | 60.9 | 35.2 | 38.2 | 46.0 | 31.4 | 45.6 |
| Spatial-MLLM-4B (Wu et al., 2025a) | 48.4 | 65.3 | 34.8 | 63.1 | 45.1 | 41.3 | 46.2 | 33.5 | 46.3 |
| ViLaSR-7B (Wu et al., 2025b) | 45.4 | 63.5 | 34.4 | 60.6 | 30.9 | 48.9 | 45.2 | 30.4 | 49.2 |
| SpatialLadder-3B (Li et al., 2025a) | 45.7 | 63.5 | 34.3 | 61.7 | 43.9 | 45.4 | 44.8 | 35.6 | 36.4 |
| VST-RL-3B (Yang et al., 2025b) | 57.7 | 66.6 | 45.0 | **72.8** | 60.9 | 59.9 | 47.6 | 40.7 | 68.3 |
| **3D-RFT-4B (ours)** | **62.8** | **71.2** | **53.5** | 70.3 | 63.2 | 60.8 | 77.9 | 37.6 | 67.8 |

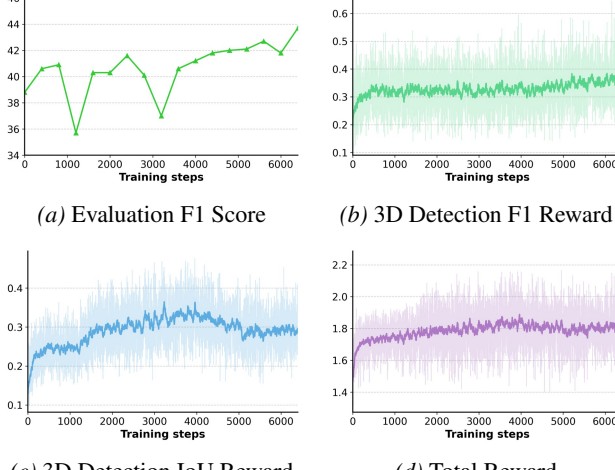

*(a)* Evaluation F1 Score     *(b)* 3D Detection F1 Reward

*(c)* 3D Detection IoU Reward     *(d)* Total Reward

*Figure 6.* **Training Dynamics Analysis (3D Video Detection).**

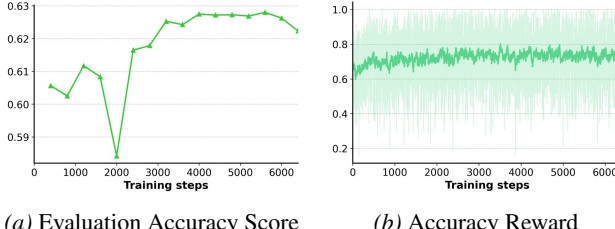

*(a)* Evaluation Accuracy Score     *(b)* Accuracy Reward

*Figure 7.* **Training Dynamics Analysis (3D Spatial Reasoning).**

Fig. 5. The results show that the SFT (DA+TA) exhibits consistently higher accuracy and rewards than SFT (DA+$\widetilde{\text{TA}}$) and SFT (TA). This suggests that the foundation of RFT relies on both DA data that teaches In-Domain (ID) knowledge and high-quality TA data (*i.e.*, CoT data) that teaches accurate reasoning behaviors.

**TA data prevents overfitting and ensures reliable reasoning behavior.** While SFT (DA) exhibits competitive In-Domain (ID) performance on VSI-Bench (Fig. 3), we observe it shows significantly poorer Out-Of-Domain (OOD) performance than SFT (DA+TA) (Sec. C.1). On the other hand, we observe that SFT (DA) can emerge reasoning output format, but the thoughts could be unreliable, *e.g.*, tedious text like R1-Zero (Guo et al., 2025). This suggests that TA data is critical for generalization and reliable reasoning, and we should avoid training MLLMs solely on DA data.

### 4.3. Training Dynamics

**3D Video Detection.** Fig. 6 illustrates the training dynamics of 3D-RFT on 3D video detection. The consistent rise in both Evaluation F1 Score (a) and F1 Reward (b) confirms that RFT effectively optimizes perception. Analyzing the reward components reveals a strategic shift: while the global F1 Reward (b) steadily increases, the IoU Reward (c) peaks early and then slightly declines. This indicates the policy transitions from initial *geometric refinement* (tightening boxes) to *recall maximization* (reducing false negatives). In the latter phase, the dense IoU Reward serves as a critical regulator, while the sparser F1 Reward optimizes the global balance between precision and recall. Moreover, the continuous improvement at the end implies that RLVR is a stable and effective paradigm for 3D perception tasks.

**3D Spatial Reasoning.** Fig. 7 illustrates the training dynamics of 3D-RFT on the spatial reasoning task. Despite minor fluctuations, both evaluation accuracy and training rewards exhibit upward trends, confirming RFT's efficacy. Notably, the curves show a sign of saturation after 4000 steps. We attribute this to the nature of the optimization

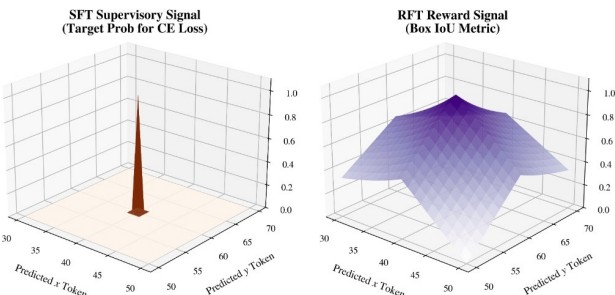

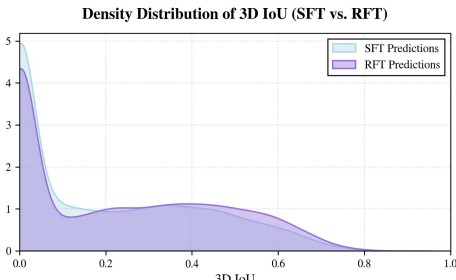

**Figure 8. Conceptual 1D Boundary Optimization Space.** We illustrate a simplified 1D proxy to illustrate the fundamental mathematical mismatch between token classification and geometric optimization. The axes represent the predicted left ($x$) and right ($y$) boundaries of a 1D interval.

**Figure 9. Distribution of 3D IoU in 3D Visual Grounding.** A comparison of the raw 3D IoU distributions on the validation set between the baseline model trained with SFT and SFT.

landscape: unlike perception tasks where RFT continuously refines continuous geometric coordinates, the reasoning task features a discrete text space with coarser feedback. This leads to the earlier saturation than 3D perception tasks, and potentially limits the granularity of improvement over SFT.

### 4.4. Analysis on Training Objectives of SFT and RFT

While Sec. 1 introduces how 3D-RFT provides a direct optimization signal compared to standard SFT in 3D scene understanding tasks, this section presents the theoretical analysis and empirical evidence to support this claim.

#### 4.4.1. THEORETICAL ANALYSIS OF SUPERVISORY SIGNALS

We model the supervisory signals for a simplified bounding box prediction task to contrast the gradient landscapes of cross-entropy and metric-driven optimization (Fig. 8).

**The SFT supervisory signal**: Standard SFT optimizes next-token likelihood via CE loss against a one-hot target. As illustrated in Fig. 8 (left), this formulation yields a sparse, binary target probability distribution—an isolated spike exactly at the ground truth. The objective lacks ordinal feedback, leaving the model blind to spatial proximity.

**The RFT reward signal**: In contrast, RFT bypasses this non-differentiable barrier by treating the final evaluation metric (e.g., IoU) as a black-box reward. This formulation constructs a smooth, continuous reward surface (Fig. 8, right). The model receives explicit feedback based on geometric overlap, which effectively guides the optimization trajectory toward high-precision regions even when initial predictions are imperfect.

#### 4.4.2. EMPIRICAL EVIDENCE IN 3D VISUAL GROUNDING

To evaluate how these theoretical differences dictate real-world performance, we analyze the probability density dis-

tributions of IoU predictions on the ScanRefer evaluation set for both the SFT baseline and the RFT model (Fig. 9).

**SFT baseline (all-or-nothing phenomona)**: The SFT distribution (Fig. 9, light blue) exhibits a severe density spike at exactly IoU = 0.0. Because token-level CE loss provides no spatial smoothing, a minor error in the token generation sequence often results in a completely dislocated 3D bounding box, leading to catastrophic failure.

**RFT advance (graceful degradation)**: By optimizing directly against the continuous evaluation metric, RFT fundamentally alters this error distribution. As shown by the purple curve in Fig. 9, RFT substantially suppresses the zero-overlap spike and shifts the probability mass to the right. This forms a prominent "spatial hump"—particularly pronounced between IoU thresholds of 0.25 and 0.50—indicating that errors are bounded to localized regions.

The empirical evidence echoes our assumption in Sec. 4.4.1 that RFT provides better optimization signals than SFT.

### 5. Conclusion

We introduce 3D-RFT, a framework that applies RLVR to video-based 3D scene understanding. By shifting the learning paradigm from token-level imitation to direct metrics-driven optimization, *i.e.*, utilizing verifiable rewards like 3D IoU and F1-Score, 3D-RFT effectively aligns training objectives with task performance. We demonstrate that 3D-RFT effectively enhances model performance across perception tasks and reasoning tasks, including 3D video detection, 3D visual grounding, and spatial reasoning. Our model 3D-RFT-4B even outperforms larger baselines like VG LLM-8B in 3D video detection and VLM-3R-7B in spatial reasoning. Our analyses reveal good properties of 3D-RFT such as robust efficacy, and valuable insights like the impact of data diversity. We hope our work can unveil the potential of RLVR for 3D scene understanding and offer valuable insights for future research.

## Impact Statement

This paper presents work whose goal is to advance the field of Machine Learning. There are many potential societal consequences of our work, none of which we feel must be specifically highlighted here.

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

# A. Implementation Details

## A.1. Data

**3D Video Detection.** For 3D video detection, we re-use the data from VG LLM (Zheng et al., 2025b), which was initially curated from EmbodiedScan (Wang et al., 2024). The data comprises consecutive frames and the corresponding visible object annotations in indoor scenes, covering 958 scenes for training and 243 scenes for evaluation. The object 3D bounding boxes are transformed to the coordinate system of the first frame.

**3D Visual Grounding.** We use the ScanRefer (Chen et al., 2020) dataset for 3D visual grounding, which is a common practice. ScanRefer includes 37K pairs of object grounding text and axis-aligned 3D bounding boxes across 562 indoor scans. We follow prior works (Zhang et al., 2025a; Zheng et al., 2025b) in terms of spatio-temporal grounding formulation and data processing procedures.

**3D Spatial Reasoning.** Our SFT data for the spatial reasoning task includes VSI-207K from VLM-3R (Fan et al., 2025) and Sr-91K from SpaceR (Ouyang et al., 2025), spanning ScanNet (Dai et al., 2017), ScanNet++ (Yeshwanth et al., 2023), and ARKitScenes (Baruch et al., 2021). Our CoT data is generated by Qwen3-VL-32B-Thinking (Bai et al., 2025a) and curated by selecting those whose accuracy rewards exceed $0.85$. The curated CoT-10K comprises 9.2K from VLM-3R-VSI subset (Fan et al., 2025) and 0.8K from SQA3D (Ma et al., 2023).

## A.2. Training

**Training Settings.** All experiments are conducted on 8 Nvidia A100 GPUs. For the 3D video detection and 3D visual grounding tasks, we adopt the full fine-tuning strategy following the settings in VG LLM. In contrast, for 3D spatial reasoning, we employ LoRA (Hu et al., 2022) and set the KL-Divergence penalty coefficient to zero during policy training.

**GPU Memory Bottleneck in 3D-RFT.** A critical bottleneck in video-based 3D scene understanding tasks is the prohibitive GPU memory consumption required by GRPO. The necessity of maintaining a large group size leads to substantial memory spikes. Specifically, processing long-context video inputs with high-resolution vision encoders (e.g., VGGT) generates extensive computation graphs. In standard implementations, storing activations for all $K$ samples simultaneously often exceeds VRAM capacity, resulting in Out-Of-Memory (OOM) errors.

**Loss Backward Chunking.** To mitigate this, we introduce a gradient accumulation strategy that partitions the global batch indices $\{1, \ldots, K\}$ into $M$ disjoint micro-chunks $\{C_1, \ldots, C_M\}$. The size of each chunk is constrained such that $|C_m| \leqslant M_{\text{micro}}$ to fit within GPU memory limits. Accordingly, we reformulate the total loss as a sum of chunked losses:

$$\mathcal{L}_{\text{chunked}}(\theta) = \sum_{m=1}^{M} \frac{|C_m|}{K} \mathcal{L}_{C_m}(\theta), \tag{15}$$

$$\text{where} \quad \mathcal{L}_{C_m}(\theta) = \frac{1}{|C_m|} \sum_{k \in C_m} \sum_{t} \mathcal{L}_{k,t}(\theta). \tag{16}$$

By iteratively computing and accumulating gradients for each $\mathcal{L}_{C_m}$, we reduce the peak memory complexity from $\mathcal{O}(B \times G)$ to $\mathcal{O}(M_{\text{micro}})$. This approach enables significant scaling of the group size $G$ without hurting the training performance. In our experiments, we set the micro-chunk size to 2 for 3D video detection, 4 for 3D visual grounding, and 1 for 3D spatial reasoning.

# B. Additional Results

## B.1. Quantitative Results

We report the evaluation results on more spatial reasoning benchmarks that adopt multi-view input, including VSI-Bench (Yang et al., 2025a), MindCube (Yin et al., 2025), MMSI-Bench (Yang et al., 2025c), All-Angles (Yeh et al., 2026), and ERQA (Team et al., 2025a). VSI-Bench adopts 32-frame inputs, while the other benchmarks adopt only several images as input. Specifically, we compare the performance of 3D-RFT to VST (Yang et al., 2025b), a generalist model trained on over 6M samples spanning single-image, multi-image, and video settings. As shown in Tab. 5, we report the following findings: (1) 3D-RFT-8B performs better than 3D-RFT-4B in general, confirming the effect of model scaling under the 3D-RFT framework. (2) 3D-RFT achieves better in-distribution performance (VSI-Bench) compared to VST,

*Table 5.* **Additional results across more spatial reasoning benchmarks and different model scales.**

|  | VSI-Bench | MindCube | MMSI-Bench | All-Angles | ERQA |
|---|---|---|---|---|---|
| VST-3B (Yang et al., 2025b) | 57.7 | 35.9 | 31.3 | - | - |
| 3D-RFT-4B (ours) | 62.8 | 36.4 | 29.9 | 41.6 | 34.2 |
| VST-7B (Yang et al., 2025b) | 61.2 | 39.7 | 34.8 | - | - |
| 3D-RFT-8B (ours) | 64.0 | 42.6 | 28.0 | 43.1 | 38.5 |

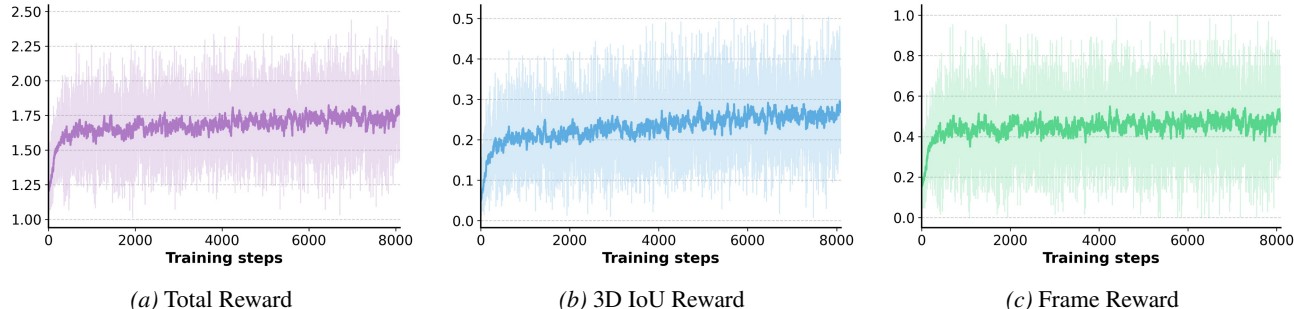

*(a)* Total Reward      *(b)* 3D IoU Reward      *(c)* Frame Reward

*Figure 10.* **Training Dynamics of 3D Visual Grounding.** We visualize the training curves for the total reward (left), the 3D IoU reward (middle), and the Frame Alignment reward (right). The curves show stable convergence and correlation with the optimization objectives.

including both 3B-4B and 7B-8B scales. (3) 3D-RFT slightly outperforms VST on MindCube and underperforms VST on MMSI-Bench. We attribute this to the substantial difference in training data: VST is trained on 6M samples spanning single-image, multi-image, and video settings, whereas our spatial reasoning model uses only 300K video-focused samples. The limited scale and diversity place our model at a disadvantage on this out-of-distribution benchmark, and also explain why performance does not scale positively from 4B to 8B on MMSI-Bench as it does on VSI-Bench. Nonetheless, our primary goal is to develop an RLVR framework for video-based 3D understanding rather than a generalist model, and we believe that the out-of-distribution performance can be aided by incorporating more diverse training data.

## B.2. Qualitative Results

### B.2.1. 3D VIDEO DETECTION

The qualitative results for 3D video detection in Fig. 12 corroborate our quantitative analysis, highlighting the enhanced precision of 3D-RFT. Consistent with the reduction in false positives reported in Tab. 1, 3D-RFT-4B successfully avoids the hallucinations generated by the baseline. For instance, in the first row, VG LLM-4B misinterprets a curtain's shadow as a valid box, and in the second row, it hallucinates a table on the floor; our model correctly suppresses both errors. Furthermore, 3D-RFT-4B demonstrates superior semantic granularity and recall: in the last row, it precisely classifies "toilet paper" (improving upon the baseline's coarser label "paper"), and in the third row, it successfully detects a "desk" that the baseline misses.

### B.2.2. 3D VISUAL GROUNDING

In Fig. 13, we provide qualitative results of 3D visual grounding. The evaluation is conducted under a 32-frame setting, and we choose some key frames mannually for visualization.

### B.2.3. 3D SPATIAL REASONING

We present qualitative results for 3D spatial reasoning in Sec. C.2. In these cases, we sample some frames from the raw video. These examples demonstrate that 3D-RFT-4B is capable of generating concise, scene-grounded CoT reasoning to facilitate accurate answer prediction.

## B.3. Training Dynamics of 3D Visual Grounding

We illustrate the training dynamics of 3D visual grounding in Fig. 10. The curves demonstrate a consistent upward trend for both the frame reward and the 3D IoU reward, indicating a stable and effective optimization process.

## C. Discussion

### C.1. Impact of TA Data for SFT

We compare the OOD evaluation results of SFT (DA) and SFT (DA+TA) on MindCube-Tiny (Yin et al., 2025). The results in Fig. 11 show that SFT (DA) significantly lags behind SFT (DA+TA) under both TA and DA test settings. SFT (DA) also shows notable performance drops compared to the vanilla Qwen2.5-VL-3B baseline. This indicates the potential overfitting issues in training 3D MLLM with solely DA data, appealing for mixing more TA data (*i.e.*, CoT data) for comprehensive enhancement of MLLM's capabilities.

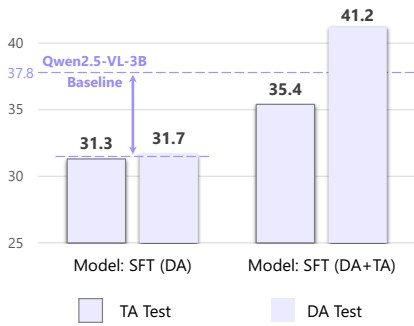

*Figure 11.* **Evaluation results on MindCube-Tiny**.

### C.2. Remarks

Unlike domains where RLVR thrives (*e.g.*, math and coding), 3D scene understanding presents distinct challenges. First, the scarcity of high-quality labels and the inherent sparsity of learning signals constitute major obstacles. The cost and difficulty of 3D CoT data collection further hinder the efficacy of RLVR for 3D scene understanding. Our findings suggest that RLVR outcomes are sensitive to CoT data quality, highlighting the refinement of CoT data as a critical future direction. Second, while math and coding tasks exhibit a straightforward reasoning loop within a unified textual modality, 3D scene understanding requires robust 3D-aware perception from video inputs before reasoning can occur. This perceptual stage represents a significant bottleneck, which could undermine the correctness and coherence of the model's thoughts. Therefore, future research should prioritize process reward designs to guarantee the soundness of reasoning within 3D scenes.

## D. Limitations

In this work, we focus on demonstrating the efficacy of the RFT paradigm by conducting task-specific fine-tuning during the RL stage; we do not currently explore unified, multi-task fine-tuning. Consequently, investigating how to balance diverse task rewards to enable efficient mixed training remains a critical direction for future research. Furthermore, we do not utilize CoT data for 3D perception tasks due to the challenges associated with collecting high-quality reasoning annotations in this domain. Acquiring such data and analyzing the impact of CoT on perception capabilities represents another promising avenue for advancing video-based 3D scene understanding models.

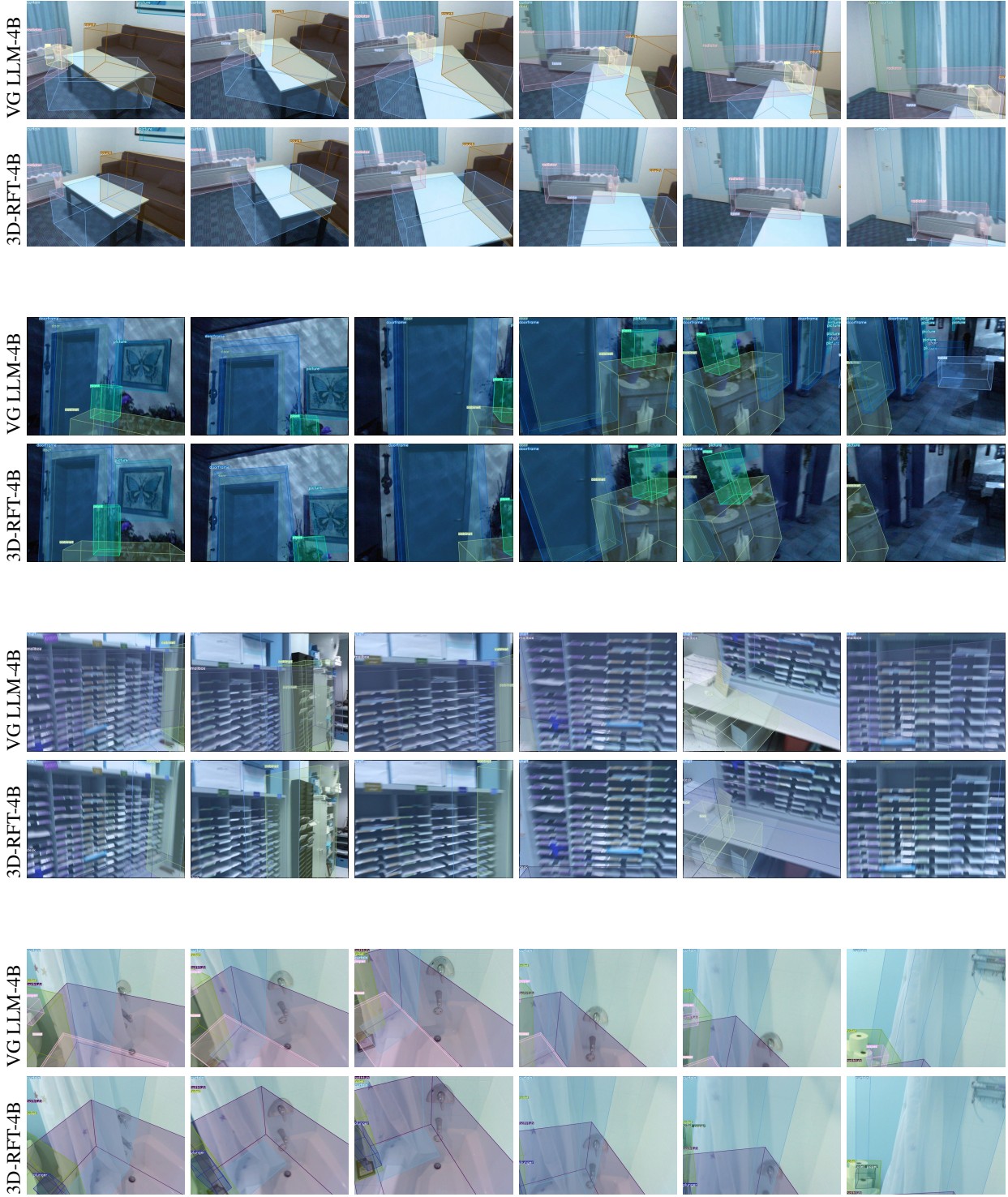

*Figure 12.* **Qualitative comparison of 3D object detection (20 common classes) results.** We compare the predictions of the baseline **VG LLM-4B** against our **3D-RFT-4B** model.

**Grounding Text:** *"There are two black chairs situated between six brown chairs and a black couch. this black chair is next to the black couch. it appears to be leather. it is black. there is a snack machine on the opposite wall."*

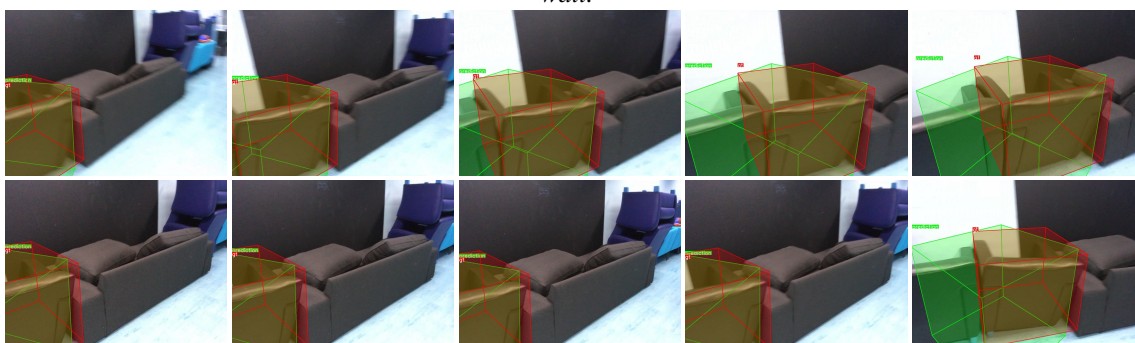

**Grounding Text:** *"A black top located in center of desk. the lid is open and blue screen is visible."*

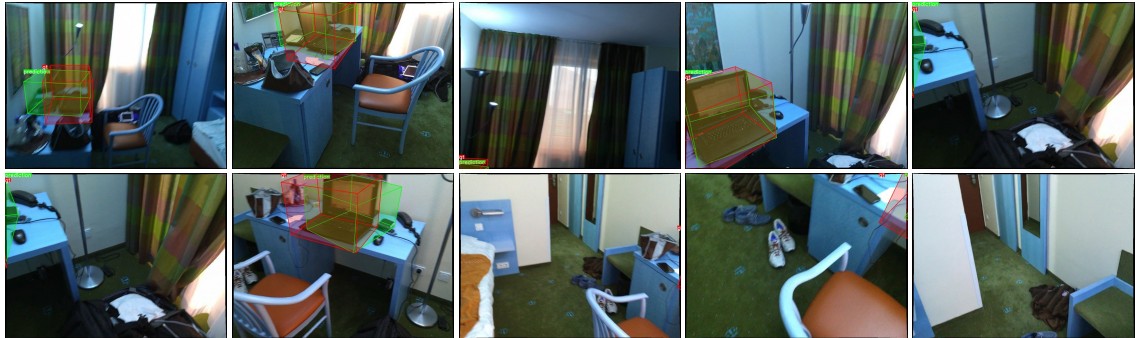

**Grounding Text:** *"The brown desk is right next to the window by the chair. the desk is also right in front of the door."*

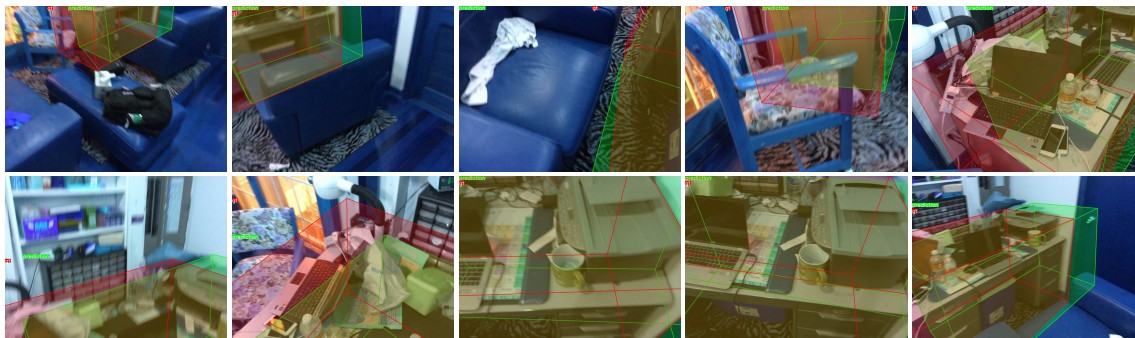

**Grounding Text:** *"This is a black table. it is to the right of the kitchen cabinet."*

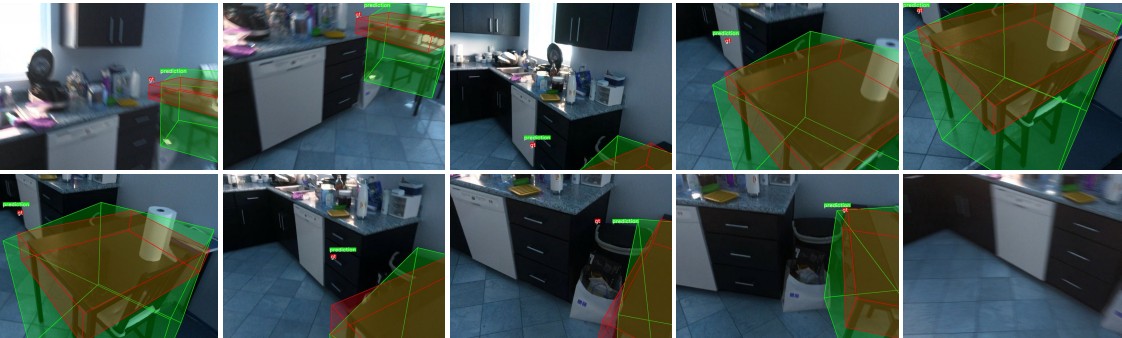

*Figure 13.* **Qualitative results of 3D Visual Grounding using our 3D-RFT-4B**. Each block shows the grounding text followed by 10 video frames visualizing the predicted 3D bounding boxes (green) and ground truth (red).

**Video**

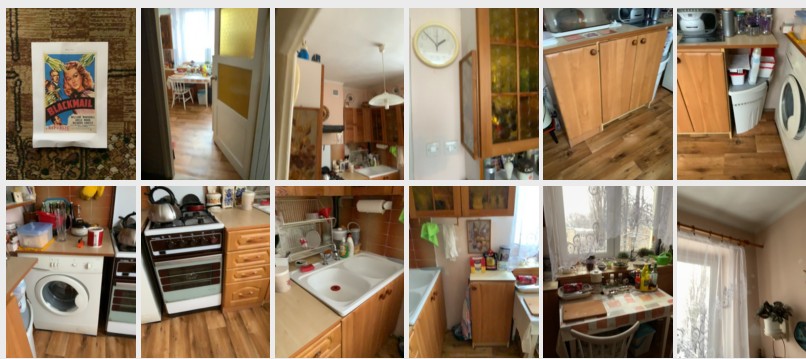

**Question**

*These are frames of a video. Measuring from the closest point of each object, what is the distance between the stool and the washer (in meters)?*

**3D-RFT-4B**

```
<think>
The stool is near the table and the washer is in the lower part of the kitchen area.
</think>
<answer>1.4</answer>
```
**GT:** 1.3

*Figure 14.* **Qualitative example on VSI-Bench (absolute distance).**

**Video**

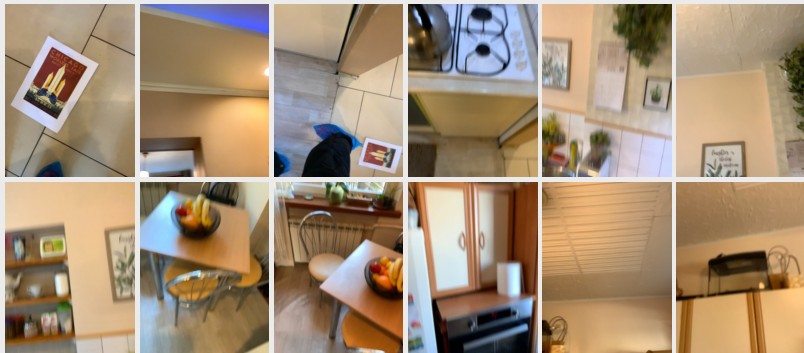

**Question**

*These are frames of a video. If I am standing by the stove and facing the refrigerator, is the table to my front-left, front-right, back-left, or back-right? The directions refer to the quadrants of a Cartesian plane (if I am standing at the origin and facing along the positive y-axis).*
*Options: A. front-right B. front-left C. back-left D. back-right*

**3D-RFT-4B**

```
<think>
I am standing by the stove, which is on the left side of the kitchen.  The
refrigerator is on the right side of the kitchen.  The table is located in the middle
of the kitchen, to the right of the refrigerator.
</think>
<answer>A</answer>
```
**GT:** A

*Figure 15.* **Qualitative example on VSI-Bench (relative direction).**

**Video**

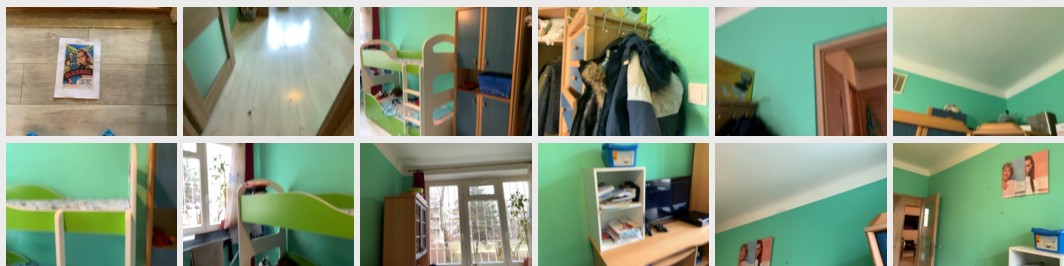

**Question**

*These are frames of a video. You are a robot beginning at the door and facing the desk. You want to navigate to the window. You will perform the following actions (Note: for each [please fill in], choose either "turn back," "turn left," or "turn right."): 1. Go forward until the desk 2. [please fill in] 3. Go forward until the bed 4. [please fill in] 5. Go forward until the window. You have reached the final destination. Options: A. Turn Right, Turn Right B. Turn Right, Turn Left C. Turn Back, Turn Right D. Turn Left, Turn Left*

**3D-RFT-4B**

```
<think>
The robot starts at the door facing the desk.  To reach the window, it must first go
forward to the desk, then turn right to face the bed, and finally turn right again to
reach the window.
</think>
<answer>B</answer>  GT: B
```

*Figure 16.* **Qualitative example on VSI-Bench (route plan).**

