# OpenReview forum: "3D-RFT: Reinforcement Fine-Tuning for Video-based 3D Scene Understanding"
_ICML.cc/2026/Conference — ICML 2026 regular_

### Official Review · Reviewer_oULb · 2026-02-13

**Soundness:** 3
**Presentation:** 3
**Significance:** 2
**Originality:** 2
**Overall Recommendation:** 4
**Confidence:** 5

**Summary:**

This paper introduces 3D-RFT. It is a framework that uses RL to video-based 3D scene understanding. It also uses SFT in this domain. The paper explores the evaluation metrics in 3D scene understanding, including 3D IoU and F1-Score. The experimental results including 3D video detection, 3D visual grounding, and 3D spatial reasoning. The results on 4B model are promising with strong performance on several benchmarks like ScanNetDetection, ScanRefer, VSI-Bench.

**Compliance With Llm Reviewing Policy:**

Affirmed.

**Final Justification:**

The authors' final reply resolve my concerns. Thus, I improve the rating to weak acceptance.

**Key Questions For Authors:**

I will improve the rating if the 7B model and some of the required benchmarks can be reported with promising performance.

**Limitations:**

Please see the weaknesses.

**Strengths And Weaknesses:**

Strengths

1. The framework is unified for multiple tasks, including 3D video detection, 3D visual grounding, and 3D spatial reasoning. This multi-task capacity is promising for a 4B model.

2. The reward design is detailed and effective, like 3D IoU, F1-Score, which ensures the performance for stable RL training process.

Weakness

1. The performance on 7B model should be reported. Please compare with VST-7B - "Visual spatial tuning Arxiv 2025" on the 7B scale.

2. Some benchmarks are ignored. Please compare on benchmarks BLINK, AlAngles, ERQA, VSI, 3DSR, MMSI, CV-3D, RealWorldQA.

3. The paper claims that this is the first framework to extend RL to video-based 3D perception / reasoning. However, VST uses RL in this domain. Thus, this is an over-claim.

4. The paper uses two separate models for 3D grounding and 3D video understanding? Why not use a unified model?

5. The novelty is the video grounding reward design. Other part of methods share novelty to existing works.

---

> ### Author Rebuttal · Authors · 2026-03-31
>
> We greatly appreciate the reviewer’s careful assessment and insightful comments. We respond to the concerns as follows.
>
> ### **W1 & W2**
>
> > 7B performance and comparison with VST-7B
>
> > More benchmarks: BLINK, AlAngles, ERQA, VSI, 3DSR, MMSI, CV-3D, RealWorldQA.
>
> We report a subset of additional benchmarks as below due to limited time and resources. We observe higher general performances in 3D-RFT-8B. While 3D-RFT is weaker than VST [1] in MMSI-Bench, we attribute this to the difference in training data. VST is trained on a comprehensive data scheme covering single-image, multi-image, and video ranges, amouting to over 6M in total. In contrast, our spatial reasoning model only involves 300k data, with much smaller scale and less diversity, which puts our work in a weak position in out-of-distribution tests. Our primary focus is an RFT framework rather than a generalist model. We will incorporate results of 3D-RFT-8B for both 3D detection and grounding in the revised manuscript.
>
> |Reasoning|VSI-Bench|MindCube|MMSI-Bench|All-Angles|ERQA|
> |-|-|-|-|-|-|
> |VST-3B|57.7|35.9|31.3|-|-|
> |3D-RFT-4B|62.8|36.4|29.9|41.6|34.2|
> |VST-7B|61.2|39.7|34.8|-|-|
> |3D-RFT-8B|64.0|42.6|28.0|43.1|38.5|
>
> ### **W3**
>
> > Over-claim on the first framework to extend RL to video-based 3D perception / reasoning
>
> We thank the reviewer's kind reminder on the technical overlap between VST [1] and our work. VST is a great work that involves RL on 3D perception and reasoning tasks. As discussed in our related work section, while VST focuses on building on a generalist model across various visual spatial tasks, we focus on the systematic study of RL on 3D perception and reasoning tasks. More specifically, we include RL studies in spatial-temporal grounding (ScanRefer), model components (3D prior), data configurations, and training dynamics. We will refine our claim to give VST more credits and emphasize our focused studies in RL.
>
> ### **W4**
>
> > Separate models for 3D grounding and 3D video understanding
>
> In this paper, we mainly focus on reward modeling on multiple video-based 3D scene understanding tasks instead of training a generalist model. This scope is consistent with Visual-RFT [2], which also focuses on reward modeling under task-specific fine-tunining settings on image-based 2D perception tasks. Our baseline VG-LLM [3] also adopts task-specific fine-tuning for perception and understanding tasks.
>
> ### **W5**
>
> > The novelty is the video grounding reward design. Other part of methods share novelty to existing works.
>
> We thank the reviewer for the thoughtful comparison to existing works like VST. While both works explore RL for spatial reasoning, they represent fundamentally different research trajectories. VST focuses on building a unified visuospatial VLM across both single and multiple images using a three-stage training pipeline (SFT $\rightarrow$ CoT-SFT $\rightarrow$ RL). In contrast, our 3D-RFT is specifically engineered for video-based 3D scene understanding, focusing on how RL provides a superior optimization signal over standard SFT in the temporal domain. Notably, our findings demonstrate that 3D-RFT achieves significant gains in perception tasks without the need for CoT data, offering unique insights into RL training dynamics. The following table summarizes these key distinctions:
>
> |  | 3D-RFT(ours) |VST|
> | -------- | -------- |-------|
> |Scope|Video-based 3D scene understanding|General understanding tasks under both single image and multiple image settings|
> | Motivation   | How does RL benifit video-based 3D scene understanding tasks beyond SFT? |How to build a strong visuospatial VLM?|
> |Settings of 3D perception tasks|3D video detection: multiple image 3D detection; 3D visual grounding: multiple image grounding; |3D detection: monocular/multiview image 3D detection
> | Main Findings  |1) RFT can benifit 3D scene understanding tasks via better optimization signal than SFT; 2) RFT can benifit 3D perception tasks without relying on COT data; 3) detailed analyses on RL training dynamics |1) carefully curated large-scale dataset and data pipeline are crucial for strong performance 2) three stage training pipeline is effective 3) the proposed RL modeling is effective|
> |Training pipeline on 3D perception tasks|SFT->RL|SFT->COT-SFT->RL|
>
> [1] Yang et al. Visual Spatial Tuning. arXiv, 2025.
>
> [2] Liu et al. Visual-RFT: Visual Reinforcement Fine-Tuning. ICCV 2025.
>
> [3] Zheng et al. Learning from Videos for 3D World: Enhancing MLLMs with 3D Vision Geometry Priors. NeurIPS 2025.

---

> > ### Author Rebuttal · Reviewer_oULb · 2026-04-02
> >
> > Thanks for the detailed reply. However, some of my concerns have not been addressed.
> >
> > For W1 & W2,
> >
> > (1) The gap to VST-7B on MMSI-Bench is a bit large, from 34.8 to 28.0.
> >
> > (2) Why 3D-RFT drops from 29.9 to 28.0 on MMSI-Bench while the model size increase from 4B to 8B?
> >
> > (3) Only 3/8 benchmarks I mentioned have been reported.
> >
> > (4) Why 3D-RFT-8B have additional 1B parameters than VST-7B? Could they be compared in the same parameter amount?
> >
> > For W3, I know the difference between 3D-RFT and VST. But claiming "the first framework" in this area is a bit tricky.

---

> > > ### Author Response · Authors · 2026-04-03
> > >
> > > Thank you for your follow-up questions. We appreciate the opportunity to clarify our evaluation scope and model design.
> > >
> > > **Q1 & Q2: Performance and Scaling on MMSI-Bench**
> > > MMSI-Bench is an out-of-distribution (OOD) task for our model. 3D-RFT (for spatial reasoning) is specialized for *video-based* 3D understanding using a focused dataset of only ~300K samples. In contrast, VST-7B is a generalist model trained on over 6 Million samples across diverse single/multi-view settings. Due to this massive difference in training scale and domain, it is expected that our model performs lower on this specific OOD benchmark and does not scale linearly (from 4B to 8B) the way it does on our target video-based tasks.
> > >
> > > **Q3: Benchmark Selection**
> > > The remaining benchmarks you mentioned (e.g., BLINK, CV-Bench, RealWorldQA) evaluate *single-image* understanding. Because our architecture and training pipeline are strictly engineered for multi-frame/video-based spatial reasoning, static single-image benchmarks fall outside the intended scope of this work.
> > >
> > > **Q4: Parameter Count (7B vs 8B)**
> > > Our framework explicitly incorporates 3D geometry priors. The 8B total comes from combining the 7B core LLM with a ~1B parameter VGGT vision-geometry encoder.
> > >
> > > **Q5 / W3: Claims and Contributions**
> > > We agree with your assessment and will remove the "first framework" phrasing in the revised manuscript. This adjustment does not affect our core scientific contributions, which focus heavily on RL mechanics rather than building a generalist model:
> > > 1. **Paradigm Shift:** Shifting the 3D perception learning objective from token-level imitation (SFT) to continuous, metrics-driven policy optimization.
> > > 2. **Reward Design:** Introducing task-specific, verifiable rewards derived directly from 3D evaluation metrics (achieving strong perception results *without* relying on CoT data).
> > > 3. **Analysis:** Demonstrating empirically and theoretically that this reward modeling overcomes the "token-level blindness" of standard Cross-Entropy loss.
> > >
> > > We hope these brief clarifications resolve your remaining concerns regarding our dataset scale and target domain. Thank you again for your time and feedback.

---

### Official Review · Reviewer_hf8D · 2026-03-12

**Soundness:** 3
**Presentation:** 3
**Significance:** 3
**Originality:** 3
**Overall Recommendation:** 4
**Confidence:** 4

**Summary:**

This paper proposes 3D-RFT, a two-stage framework for video-based 3D scene understanding. The idea is straightforward: first do SFT warm-up, then apply GRPO-based reinforcement fine-tuning with verifiable rewards that are derived directly from the target metrics. The framework covers three task types: 3D video detection, 3D visual grounding, and 3D spatial reasoning. Concretely, the reward design mirrors the evaluation protocol quite closely: detection uses 3D IoU and F1, grounding uses frame alignment plus 3D IoU, and reasoning uses exact-match / relative numerical accuracy depending on question type.

**Compliance With Llm Reviewing Policy:**

Affirmed.

**Final Justification:**

All my concerns are addressed, so I maintain my score.

**Key Questions For Authors:**

see weakness

**Limitations:**

yes

**Strengths And Weaknesses:**

- Strength

The paper starts from a clear and reasonable observation: standard SFT optimizes token-level likelihood, which is only an indirect proxy for the actual evaluation metrics in 3D detection, grounding, and reasoning. 3D-RFT addresses this by using task-specific verifiable rewards derived directly from metrics such as 3D IoU, F1, and answer accuracy, which is a natural and meaningful reformulation of the training objective.

One appealing aspect of the paper is that the same overall RL fine-tuning framework is applied to three fairly different settings: 3D video detection, 3D visual grounding, and 3D spatial reasoning. The paper also enforces structured outputs and combines format rewards with task rewards, which makes the framework coherent rather than task-by-task improvised.

The paper does more than just report final numbers. It studies the effect of 3D priors, shows that RFT helps both with and without VGGT input, and analyzes the role of DA/TA data and CoT quality for spatial reasoning. These ablations make the paper more informative than a pure benchmark-improvement submission.

- Weakness
The reasoning setup relies on a large SFT mixture plus an additional high-quality CoT-10K set generated by Qwen3-VL-32B-Thinking, and the paper itself shows that performance is sensitive to DA/TA composition and CoT quality. This makes it harder to isolate how much of the final improvement comes from RL itself, as opposed to stronger reasoning supervision and data curation.

- The method is introduced as a KL-regularized GRPO framework with a frozen reference model, but in the appendix, the spatial reasoning setting switches to LoRA and sets the KL-divergence penalty coefficient to zero during policy training. This makes the training recipe feel more task-specific than the main framing implies, and weakens the claim of a single unified RL formulation across all tasks.

---

> ### Author Rebuttal · Authors · 2026-03-31
>
> We greatly appreciate the reviewer’s careful assessment and insightful comments. We respond to the concerns as follows.
>
> ### **W1**
>
> > The reasoning setup relies on a large SFT mixture plus an additional high-quality CoT-10K set generated by Qwen3-VL-32B-Thinking, and the paper itself shows that performance is sensitive to DA/TA composition and CoT quality. This makes it harder to isolate how much of the final improvement comes from RL itself, as opposed to stronger reasoning supervision and data curation.
>
> Thanks for the reviewer's rigorous comments. We clarify that the results of improvements from RL are presented in Figure 3, which directly compares the performances before and after RL. As CoT (TA) data is a necessary prerequisite for establishing the reasoning format for RLVR, it has been considered an inherent part of RL since prior work [1]. Our additional analyses on "DA/TA composition and CoT quality" intend to offer deeper insights into the key factors that modulate RL efficacy.
>
> Furthermore, recent studies such as VST [2] have reported a similar phenomenon: RL tends to yield smaller improvements in spatial reasoning compared to perception tasks (see Tables 2 and 4 in [2]). Our hypothesis for this observation is that the discrepancy between the evaluation metric and the token-level cross-entropy (CE) loss is significantly smaller in spatial reasoning than in 3D perception.
>
> For instance, in 3D detection, the mapping from predicted 9-DoF text coordinates to the final F1-score is much more indirect than the mapping from text output to accuracy in a reasoning task. This inherent complexity in the perception reward landscape makes the optimization benefits of RL more pronounced in those tasks.
>
> ### **W2**
>
> > The method is introduced as a KL-regularized GRPO framework with a frozen reference model, but in the appendix, the spatial reasoning setting switches to LoRA and sets the KL-divergence penalty coefficient to zero during policy training. This makes the training recipe feel more task-specific than the main framing implies, and weakens the claim of a single unified RL formulation across all tasks.
>
> We thank the reviewer for this insightful observation. While 3D-RFT provides a unified conceptual framework, our specific implementation choices (e.g., the use of LoRA vs. full fine-tuning or the application of KL penalties) are deliberately tailored to the inherent technical demands of each task.
>
> - **Task-Specific Challenges**: Perception tasks require the model to regress precise, continuous 9-DoF coordinates—a high-precision requirement that is notoriously challenging for LLMs compared to language-based reasoning. To address this difficulty, we adopt full fine-tuning for perception, a well-established practice in state-of-the-art benchmarks [3, 4, 5].
> - **Methodological Focus**: As our primary objective is to investigate how Reinforcement Learning (RL) benefits video-based 3D scene understanding in multi-image settings and how to design effective rewards, we follow the established methodology of Visual-RFT [6] by employing task-specific fine-tuning.
>
>
> [1] Guo et al. DeepSeek-R1: Incentivizing Reasoning Capability in LLMs via Reinforcement Learning. arXiv, 2025.
>
> [2] Yang et al. Visual Spatial Tuning. arxiv, 2025.
>
> [3] Chen et al. LL3DA: Visual Interactive Instruction Tuning for Omni-3D Understanding, Reasoning, and Planning. CVPR 2024.
>
> [4] Zhu et al. LLaVA-3D: A Simple yet Effective Pathway to Empowering LMMs with 3D-awareness. ICCV 2025.
>
> [5] Zheng et al. Learning from Videos for 3D World: Enhancing MLLMs with 3D Vision Geometry Priors. NeurIPS 2025.
>
> [6] Liu et al. Visual-RFT: Visual Reinforcement Fine-Tuning. ICCV 2025
>
> Thank you again for your constructive and valuable suggestions. If you have any further concerns, please do not hesitate to let us know.

---

> > ### Author Rebuttal · Reviewer_hf8D · 2026-04-01
> >
> > All my concerns have been addressed, so I maintain my score to weak accept.

---

> > > ### Author Response · Authors · 2026-04-03
> > >
> > > Dear Reviewer hf8D,
> > >
> > > Thank you for your prompt acknowledgement and for confirming that our rebuttal has fully addressed your concerns! We deeply appreciate the time you took to evaluate our work and your continued support for our submission.
> > >
> > > Because you specifically highlighted the strength of our verifiable reward design in your initial review, we wanted to briefly bring to your attention some new analyses we just posted in response to another reviewer's (Reviewer Dtde) follow-up questions.
> > >
> > > To provide even stronger mathematical and empirical proof of our method's advantages over standard SFT, we have added:
> > >
> > > **Theoretical Analysis**: We modeled the supervisory signals for 2D bounding box prediction. The visualization clearly shows how standard SFT creates a sparse, binary target (token-level blindness), whereas RFT bypasses the non-differentiable barrier to create a smooth, continuous reward surface that safely guides the optimization trajectory.
> > >
> > > **Empirical Evidence**: We plotted the probability density distributions of IoU predictions on the ScanRefer test set. The data proves our theoretical claim: RFT successfully shifts the probability mass away from the catastrophic zero-overlap failures seen in SFT and seamlessly degrades into geometric "near-misses."
> > >
> > > We believe these additions make the fundamental advantage of 3D-RFT's continuous geometric awareness undeniable.
> > >
> > > If you feel that these new theoretical and empirical analyses further strengthen the manuscript, we would be incredibly grateful if you might consider reflecting this in your final evaluation. Regardless, we hope these additions give you even greater confidence in the framework, and any advocacy you might provide for the paper's strengths during the upcoming discussion phase would be deeply appreciated.
> > >
> > > Thank you again for being such an engaged and constructive reviewer!

---

### Official Review · Reviewer_Dtde · 2026-03-13

**Soundness:** 3
**Presentation:** 3
**Significance:** 2
**Originality:** 3
**Overall Recommendation:** 4
**Confidence:** 3

**Summary:**

The paper's tasks covered 3D object detection, 3D grounding and 3D spatial reasoning, and they claims that SFT's token-level CE loss and the training objective is mismatched, so they uses RL to directly optimize for these target metrics. The training recipe is SFT with RL (GRPO). And results show that their methodology is effective.

**Compliance With Llm Reviewing Policy:**

Affirmed.

**Final Justification:**

Thank you. Please consider incorporating the rebuttal into the final version. I have raised my score.

**Key Questions For Authors:**

- Line 232 right: Did the indicator function introduce training instabilities? it seems no ablation studies on reward design choices.
- How does the method compare to traditional (non-LLM/MLLM) video understanding models that are more specialized for this domain?
- Figure 7: why there's a obvious drop in evaluation accuracy step 2000, is this just *minor fluctuations* and can be ignored?
- what is the memory usage, framework used. how many K of candidates in GRPO? also does the G in Line 136 right is the same as K Line 797?
- The figures are pleasing to eyes, but better consider reducing font size?

**Limitations:**

yes

**Strengths And Weaknesses:**

Strengths:
- the paper is easy to follow.
- good results, outperforming other methods across multiple tasks.
- Figs and Tables are pleasing to eyes.
- successful domain-specific adaptation.

Weaknesses:
- my biggest concern is that the authors seem to have bias on the SFT-only training and, in theory, I hold different views against: abstract Line 17-20, right column Line 31-45, which are basically the motivation why the authors use RFT to optimize the training. Those claims seem to be too strong. I recommend the authors check out this second best paper of neurips2025: *Does Reinforcement Learning Really Incentivize Reasoning Capacity in LLMs Beyond the Base Model?* they use convincing evidence (pass@k, perplexity, etc) to show that RL's ability is bound by the SFT model. therefore, the base SFT model can actually perform good as RL does, though many attempts may be required.  **In contrast, in this paper, the authors say the *token level cross-entropy loss* is the root cause for the performance *ceiling*. (so RFT is required)** if the authors stick to this statement, more convincing evidence should be presented.
- Furthermore, SFT can also incorporate metric-based terms, the loss function could be redesigned to include task-specific supervision signals beyond CE loss.
- The benefit of reward functions aligned with task-specific targets is not unsurprising. im afraid this barely brings new insights to the broader AI community.

---

> ### Author Rebuttal · Authors · 2026-03-31
>
> We greatly appreciate the reviewer’s careful assessment and insightful comments. We respond to the concerns as follows.
>
> ### **W1**
> > Comparison of SFT and RFT. Motivation of RFT.
>
>
> **The Scope of RL:**
>
> We thank the reviewer for pointing us toward the Limit-of-RLVR paper [1]. While we have discussed this work in our paper, here we clarify the distinction between linguistic reasoning trajectories and the high-precision numerical output required for 3D perception.
>
> Reasoning (CoT): The cited paper suggests RL acts as a "path-finder" through logical steps exists in the SFT distribution. While RL simply optimizes the selection of existing knowledge.
>
> 3D Perception (Ours): In 3D perception tasks, we isolate RL's role and explore how it provide better numerical calibration, while COT reasoning is not our focus. Standard SFT treats numerical tokens (e.g., "0", ".", "5") as discrete symbols. Consequently, CE loss cannot distinguish between a "near-miss" (predicting 0.51 instead of 0.50) and a "far-miss" (0.99 instead of 0.50), as it weights all incorrect tokens equally [2]. Furthermore, we do not consider COT reasoning for perception tasks. Thus, the sampling space of the model output is much smaller. More sampling of SFT model may be insufficient for better performance.
>
> Conclusion: Our "ceiling" claim refers to this Token-Level Blindness [3].In 3D perception, our approach overcomes the inherent inability of CE loss to achieve the fine-grained spatial precision required for 3D tasks.
>
>
> ### **W2 & W3**
> > SFT can also incorporate metric-based terms
>
> > Unsurprising benefit of RFT
>
> **Why Redesigning SFT Loss is Intractable:**
>
> The reviewer suggests incorporating metric-based terms into the SFT loss. However, standard SFT relies on backpropagation, which **requires a fully differentiable path from the loss function to the model’s logits**. Using target 3D metrics as an SFT loss is **mathematically intractable since the evaluation pipeline involves discrete, non-differentiable operations**. Specifically, converting text to a 3D box requires **string parsing, and calculating the final metric involves step functions** (e.g., $\text{IoU} \geq 0.25$). If these were added to the SFT loss, they would **produce zero or undefined gradients**, halting the learning process. While differentiable proxies (such as Smooth-L1 loss) often misalign with the true metrics.
>
> **The RL Advantage:**
>
> RL (via GRPO) bypasses this by treating the evaluation pipeline as a "black-box" reward. **Using the Policy Gradient theorem [4], we can directly optimize for discrete target metrics without needing a differentiable loss landscape**.
>
>
> ### **Q1**
>
>
> > Does indicator function introduce training instabilities?
>
> **Indicator Function Stability**:
>
> The use of an indicator function as a reward signal does not introduce training instability in GRPO. Unlike SFT, RL does not require the reward function to be differentiable. The relative ranking of samples provides a sufficient signal for policy improvement.
>
> ### **Q2**
>
>
> > How does the method compare to traditional models?
>
> What we propose is a learning framework, independent of model. For traditional models, we can also calculate the rewards and employ RFT to train those models. RFT integrates output probability and rewards. The only difference across models is the output probability, e.g., MLLM uses token logits while traditional models use objectness scores.
>
> ### **Q3**
>
>
> > Evaluation accuracy drop
>
> The drop at step 2000 is a localized fluctuation commonly seen in on-policy RL as the model explores new regions of the parameter space [5]. A similar phenomenon was observed in VLM-R1 [6]. As shown in Figure 5, the evaluation performance on in-domain test data drops at the 200-step mark before recovering and increasing.
>
> ### **Q4**
>
> > Memory usage; candidates K; G  vs. K
>
> Memory usage is not a burden due to our optimization such as chunk-forward trick. For example, it costs 50GB for 3D-RFT-4B on the reasoning task. K is the global batch size (batch size $\times$ group size), G is the group size. When batch size is 1, K equals G.
>
> ### **Q5**
>
> > Font size in figures
>
> Thanks for your advice. We will refine the figures in revision.
>
> Thank you again for your constructive feedback. We hope these explanations address your concerns and would appreciate it if you would consider raising the score accordingly. We remain open to further discussion.
>
> [1] Yue et al. Does Reinforcement Learning Really Incentivize Reasoning Capacity in LLMs Beyond the Base Model? NeurIPS 2025.
>
> [2] Chen et al. Pix2Seq: A Language Modeling Framework for Object Detection. ICLR 2022.
>
> [3] Kim et al. Sequence-Level Knowledge Distillation. EMNLP 2016.
>
> [4] Sutton et al. Policy Gradient Methods for Reinforcement Learning with Function Approximation. NeurIPS 1999.
>
> [5] Schulman et al. Proximal Policy Optimization Algorithms. arXiv, 2017.
>
> [6] Shen et al. VLM-R1: A Stable and Generalizable R1-style Large Vision-Language Model. arXiv, 2025.

---

> > ### Author Rebuttal · Reviewer_Dtde · 2026-04-01
> >
> > 1.
> > > CE loss cannot distinguish between a "near-miss" (predicting 0.51 instead of 0.50) and a "far-miss" (0.99 instead of 0.50), as it weights all incorrect tokens equally [2].
> >
> > [2] in 2022 seems not solid evidence. In your setting, do SFT and RFT actually show different token probability distributions that support this claim?
> >
> > 2. Q4 is about key training details. But it seems these are not specified in the paper.

---

> > > ### Author Response · Authors · 2026-04-03
> > >
> > > Reviewer Dtde,
> > >
> > > We thank the reviewer for engaging with our rebuttal and provide further feedback. We address the follow-up questions below.
> > >
> > > ### **Q1:** **Theoretical Analysis vs. Empirical Evidence: Overcoming "Token-Level Blindness"**
> > >
> > > We thank the reviewer for pushing us to provide stronger grounding for our claims regarding standard SFT limitations. To provide the "solid evidence" requested, we present both a theoretical analysis and an empirical analysis of the models' prediction distributions (visualizations for both will be included in the revised Appendix).
> > >
> > > **Part A: Theoretical Analysis**
> > >
> > > One of the core limitations of standard SFT for high-precision perception is Metric Mismatch: its token-level objective is fundamentally disconnected from continuous spatial geometry. To illustrate this, we modeled the supervisory signals for a simplified 2D bounding box prediction $(x, y)$ (anonymous link: [[Theoretical Analysis ]](https://ibb.co/Y7cVxdmK)).
> > > - **The SFT Supervisory Signal**: Standard SFT optimizes next-token likelihood via Cross-Entropy (CE) loss against a one-hot target. As shown in the left panel, this creates a sparse, binary target probability. The signal is a single, isolated spike exactly at the ground truth. A geometric near-miss (e.g., predicting box coordinates 39, 61 instead of the ground truth 40, 60) receives the exact same zero-probability target as a far-miss. The model receives no ordinal feedback indicating it was spatially close.
> > > - **The RFT Reward Signal**: RFT bypasses the non-differentiable barrier by treating the final evaluation metric (IoU) as a "black-box" reward. As shown in the right panel, this creates a smooth, continuous reward surface. The model is explicitly rewarded based on geometric overlap, safely guiding the optimization trajectory toward precision even when the initial prediction is imperfect.
> > >
> > > **Part B: Empirical Evidence in 3D Grounding**
> > >
> > > This theoretical limitation directly dictates real-world performance. To prove this, we plotted the probability density distributions of the Intersection over Union (IoU) predictions for both SFT and RFT models on the evaluation set of 3D visual grounding (ScanRefer) (anonymous link: [Empirical Evidence in 3D Grounding](https://ibb.co/3yML8bT7)).
> > >
> > > * **SFT Baseline (All-or-Nothing):** The SFT distribution (light blue) exhibits a massive density spike at exactly `IoU = 0.0`. Because CE loss provides no spatial safety net, an imperfect token sequence generation often results in a completely dislocated 3D bounding box (a catastrophic failure).
> > > * **RFT Advance (Graceful Degradation):** By optimizing directly against the continuous evaluation metric, RFT fundamentally alters the distribution. As shown in the purple curve, RFT substantially reduces the zero-overlap spike and shifts the probability mass to the right, forming a significantly thicker "spatial hump" (particularly visible between IoUs of $0.3$ and $0.7$).
> > >
> > > **Conclusion:**
> > > Together, this evidence proves our core claim. Standard SFT is mathematically blind to metric distance, creating an artificial performance ceiling. RFT's evaluation-driven reward explicitly instills continuous geometric awareness, allowing the model to recover from token-level uncertainty and seamlessly degrade into geometric "near-misses" rather than structural failures.
> > >
> > > ### **Q2: Implementations**
> > >
> > > Our implementations (*e.g.*, group size, batch size, chunk size) follow standard practices (see [TRL documentation](https://huggingface.co/docs/trl/grpo_trainer#trl.GRPOConfig)), with minor adjustments made to accommodate available hardware resources. While these hyperparameters are not central to our framework's core contributions, we will ensure they are fully detailed in revision.
> > >
> > > We hope these further clarifications and empirical analyses fully address your remaining concerns. If you find our responses satisfactory, we would greatly appreciate it if you would consider raising your score.

---

### Decision · Program_Chairs · 2026-04-30

**Decision:**

Accept (regular)

**Comment:**

The paper proposes 3D-RFT, a method that leverages RL with verifiable rewards to improve 3D scene understanding. It received three reviews, with major concerns including unclear attribution of component contributions, task-specific design choices, limited evaluation benchmarks, and some overclaiming.
The rebuttal addressed the majority of these concerns, and all reviewers updated their scores to weak accept. The AC recommends acceptance, with the expectation that the authors incorporate all required revisions discussed during the rebuttal phase.